# A Deep Learning Model of Mental Rotation
# Informed by Interactive VR Experiments

**Raymond Khazoum** [1]  **Daniela Fernandes** [1]  **Aleksandr Krylov** [1]  **Qin Li** [2]  **Stéphane Deny** [1 2]

## Abstract

Mental rotation—the ability to compare objects seen from different viewpoints—is a fundamental example of mental simulation and spatial world modeling in humans. Here we propose a mechanistic model of human mental rotation, leveraging recent advances in deep, equivariant, and neuro-symbolic learning. Our model consists of three stacked components: (1) *an equivariant neural encoder*, producing 3D spatial representations of objects from images, (2) *a neuro-symbolic object encoder*, deriving symbolic objects descriptions from these spatial representations, and (3) *a neural decision agent*, comparing these symbolic descriptions to prescribe rotation simulations in 3D latent space *via* a recurrent pathway. Our model design is guided by the existing experimental literature on mental rotation, which we complemented with experiments in VR where participants could at times manipulate the objects to compare. Our model captures well the performance, response times and behavior of participants in our and others' experiments, and through ablation studies we demonstrate the necessity of each component. Our work adds to a recent collection of deep neural models of human spatial reasoning, further demonstrating the potency of integrating deep, equivariant, and symbolic representations to model the human mind.

> *What I cannot create, I do not understand.*
> — R. Feynman

[1]Department of Computer Science, Aalto University, Espoo, Finland [2]Department of Neuroscience and Biomedical Engineering, Aalto University, Espoo, Finland. Correspondence to: Raymond Khazoum <raymond.khazoum@aalto.fi>, Daniela Fernandes <daniela.dasilvafernandes@aalto.fi>, Aleksandr Krylov <aleksandr.krylov@aalto.fi>, Qin Li <qin.li@rwth-aachen.de>, Stéphane Deny <stephane.deny.pro@gmail.com>.

*Proceedings of the $43^{rd}$ International Conference on Machine Learning*, Seoul, South Korea. PMLR 306, 2026. Copyright 2026 by the author(s).

## 1. Introduction

Mental rotation is the human ability to compare and assess the similarity of objects seen from different viewpoints. In the early 1970s, Shepard & Metzler (1971); Metzler & Shepard (1974) systematically studied this ability through a task in which participants were asked to decide whether two images of three-dimensional objects made of cubes (hereafter, Shepard-Metzler objects), shown rotated at different angles, corresponded to the same object or mirror versions of the same object. Subjects performed the task with high accuracy, confirming their ability to mentally rotate such unfamiliar objects, but the key findings pertained to their underlying cognitive process: (1) the response time was on average proportional to the angular disparity between the objects to compare, and (2) rotating an object in depth took no longer than rotating it within the picture plane. These remarkable findings led psychologists to speculate that humans are equipped with an internal spatial engine, allowing them to manipulate representations of visual objects in their 'mind's eye'. While modern computer vision systems achieve strong performance across a wide range of 3D visual tasks, to our knowledge, no model has been proposed that would not only solve the mental rotation task, but also offer a mechanistic account of how the brain could carry out mental rotation.

In this work, we propose *a first mechanistic model of human mental rotation*. Our model draws on recent advances in deep learning, taking inspiration in particular from methods to infer 3D representations from the observation of objects seen from one or only a few viewpoints (Dupont et al., 2020; Connor et al., 2024; Spiegl et al., 2025; O'Connell et al., 2025), transformer-based image-to-sequence approaches (Vaswani et al., 2017; Dosovitskiy et al., 2021; Liu et al., 2021; Li et al., 2023), and agents performing latent simulations (Hamrick, 2019). Specifically, our model consists of three stacked modules: (1) an equivariant encoder that produces a 3D manipulable representation of an object from a single 2D view; (2) an attention-based vision encoder-decoder that produces a sequential and symbolic description of a Shepard-Metzler object from its spatial representation; and (3) a multi-layer perceptron (MLP) trained to predict

Code and Data available at: github.com/rkhz/menrot

transformational relationships from pairs of symbolic representations, deployed as an agentic system that makes similarity judgments and selects rotation actions to apply on the 3D representation—iteratively aligning the objects.

The design of our model is guided by the existing literature on mental rotation, which we complemented with a novel set of experiments in Virtual Reality (VR). While the task remained the same as originally proposed by Shepard & Metzler (1971) (comparing two Shepard-Metzler shapes), in our VR experiments subjects were able to manipulate (i.e., rotate) objects with the help of a joystick. Their actions afforded us with *a window into their mental process*, further informing our modeling effort. In particular, we found that subjects tend to take only a few ballistic actions (typically one), which reliably place the two objects in the same rough position or 'quadrant', before making a similarity judgment. We could only reproduce this efficient trial-and-error process by equipping our model with a neuro-symbolic object description engine guiding rotation decisions.

We validate our model by showing that it replicates both the performance of humans on the mental rotation task as well as the behavioral patterns observed in the literature and in our interactive VR experiments. Through systematic ablations, we demonstrate the necessity of each component of our model. To our knowledge, our work is the first attempt at a mechanistic model of human mental rotation. While fundamental questions remain, we argue that our modeling effort offers new insights into the cognitive process of mental rotation (see Sec. 6. Discussion). Furthermore, this work could inspire new approaches to computer vision and, in particular, the development of more robust and general architectures for spatial reasoning tasks.

## 2. Mental Rotation In the Literature

Our modeling effort and experimental design both build upon the extensive literature on mental rotation. Here we briefly review the relevant literature and describe how it guided our modeling and experimental choices.

**Discrete vs. Continuous Rotations.** The early finding of Shepard & Metzler (1971); Metzler & Shepard (1974) that respondents' reaction time is, on average, proportional to the angular disparity between the objects to compare, led them to posit that mental rotation is a continuous (i.e., analog) process occurring at an estimated rotation speed of 60°/s. It was however not excluded that mental rotation could be carried in discrete but regular chunks (Cooper & Shepard, 1973). Later experiments by Cooper (1976) provided the strongest psychophysical evidence in favor of a continuous—or close to continuous—process: They asked subjects to visualize a given random 2D polygon rotating progressively and at a desired speed. At some point

during this mental visualization exercise, a test shape was presented. They observed that reaction time was fast and constant when the test shape appeared in the orientation anticipated by the internal representation, but proportional to the angular difference when the test shape appeared in an unexpected orientation.[1] Complementing this psychophysical evidence in favor of continuous rotation, neuroimaging and neurostimulation studies (Zacks, 2008, for a review) suggested that brain regions involved in motor planning (e.g., SMA) could be involved in some mental rotation tasks, and observed graded responses in those regions which could correspond to a continuous rotation process. However, for other mental rotation tasks (e.g., letters), Kung & Hamm (2010); Searle & Hamm (2017) suggested that psychometric curves favor a hybrid model of continuous and discrete rotations—a hypothesis that our VR experiments also support. *Consistently with these findings and our own experimental results, we designed a model allowing for both continuous and discrete rotations.*

**Symbolic vs. Spatial Representations.** Whether the rotation is applied to a structurally isomorphic spatial representation of the shape, an abstract symbolic description, or some representation in between, is still a subject of debate (Cooper & Shepard, 1978; Pylyshyn, 2002; Kosslyn et al., 2006; Biederman, 2013, p. 124). Metzler and Shepard's observation that the mental rotation of Shepard-Metzler shapes produces the same linear reaction times whether performed in the viewing plane or in depth, at least suggests that a 3D model of the object is built, whether symbolic or abstract. Just & Carpenter (1976)'s eye-tracking experiments suggest that subjects encode Shepard-Metzler shapes sequentially. While this is not proof of a symbolic representation, Just and Carpenter and others (e.g., Levin, 1973) proposed compositional models of mental rotation where objects would be decomposed and rotated part by part. Apparently conflicting findings by Cooper & Podgorny (1976) showed that humans are able to carry out 2D mental rotation tasks in constant time regardless of the complexity of polygons, suggesting that mental rotation acts on a faithful and arbitrarily complex representation of objects. *Our model, composed of both a structurally isomorphic spatial encoder (equivariant in* SO(3)*), and of a symbolic encoder (to guide actions), is compatible and may reconcile all these conflicting accounts.*

**Further Reading.** For additional literature review on mental rotation, such as the existence of common processes between mental and manual rotation, or on the use of VR as an alternative experimental paradigm to 2D images, see App. A.

---

[1] While this result suggests the capacity of the human mind to perform continuous rotations, it does not preclude the possibility that the human mind would also be capable of discrete, larger jumps in other mental rotation tasks.

## 3. VR Experiments

We designed a novel set of experiments in Virtual Reality (VR). While Shepard & Metzler (1971) measured only reaction times on fixed images of objects, we aimed to go further by using a VR environment, which, in addition to being immersive, made it possible to design experiments utilizing controllers. Specifically, we incorporated thumbstick interactions, allowing users to manipulate objects directly. By observing their gestures, we gained access to a channel of information that provided us with additional insights into the cognitive processes of mental rotation.

### 3.1. Experimental Design

**Experiment.** Nineteen human subjects (14 males and 5 females) participated in the study. In a VR environment, subjects viewed randomly sampled pairs of 3D Shepard–Metzler shapes, each shape composed of 10 adjacent cubes forming a structure with three elbows. The shapes are presented at 25° elevation, upright as if they were sitting on the floor, with random azimuthal orientations around the Y-axis, corresponding to in-depth rotation; within each pair, objects differed by discrete relative azimuthal rotations of 0°, 60°, 120°, or 180°. After the pair is presented, the subject must quickly assess the similarity of the objects by pressing either the 'match button' (similar shapes) or the 'mismatch button' (mirror shapes). The time it takes to press the selected button is recorded as the response time. Four participants were removed from the study for atypical behavior (see App. C).

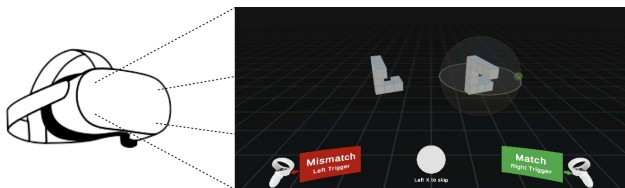

*Figure 1.* Screenshot of the VR app for the Action setup.

**Operating Modes.** The app presents two operating modes, or 'setups'. In the **No-Action setup**, corresponding to traditional mental rotation experiments, the user cannot manipulate the shapes' orientations and must perform the task purely mentally. In the **Action setup**, the user may manipulate the shape on the right with the help of a thumbstick. The shape can only rotate along the Y-axis, corresponding to the rotation axis between the shapes. During the rotation, the object disappears from view, preventing participants from seeing intermediate poses, and only an annulus around the object remains visible, with a small ball traveling along this annulus to indicate the angle of rotation (Fig. 1). The actions of subjects on the thumbstick are measured and timed. In all analyses of reaction times presented below, we only report the trials where the shapes were matching and for which the

subject correctly assessed the match (*successful matches*). Mismatch trials and incorrect trials were discarded from analyses. Further details regarding the experiments and preprocessing of the data are given in App. C.

### 3.2. Behavioral Results and their Interpretation

**Hallmark signatures of mental rotation are present in both the Action and No-Action setup, indicating shared cognitive processes between the two setups.** In line with previous reports, subjects achieved a high accuracy on the mental rotation task in both the Action and No-Action setups (App. Table 2). We observe a slight improvement in accuracy in the Action Setup. The average response time of subjects was also comparable across the two setups. Examining response times as a function of angular disparity revealed the usual linear trend in both the Action and No-Action setups (Fig. 2.A). This observation suggests that the Action and No-Action setups share common cognitive processes, allowing us to learn about mental rotation processes by studying the Action setup. This hypothesis is also consistent with prior work (see paragraph 'Mental and Manual Rotation' in App. A). We note, however, that in the Action setup the response times are not markedly different at 180° disparity than at 120° disparity (which we will comment on in the Section 5. Results).

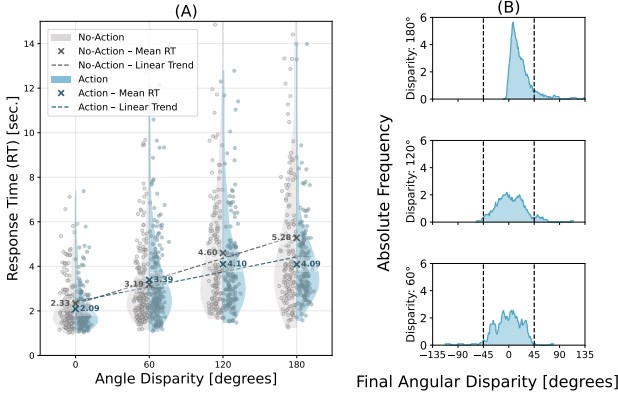

*Figure 2.* (A) Response times as a function of angular disparity, for the Action (blue) and No-Action (grey) setups. Each point represents one (successful match) trial. (B) Residual angular disparity between rotated and target objects following the last action for 180° (top), 120° (middle) and 60° (bottom) conditions.

**A number of observations converge to indicate that mental rotation allows for discontinuous jumps in rotation angle.** In both setups, we note a remarkable variability in response times across trials, for all angular disparities. Such variability, although not reported in the early studies of mental rotation, can also be found in other open datasets of mental rotation (App. Fig. 11). The fast response times sometimes observed at high angular disparity seem hardly compatible with the hypothesis of a fixed and continuous

mental rotation speed, and rather suggest that humans are able to test rotation hypotheses through discrete jumps in rotation angle. As further evidence in favor of discrete jumps, in the Action setup, subjects typically perform a few ballistic actions corresponding to large angular jumps, with a single action corresponding to an average rotation angle of 73.1°. Note that the object disappears during the rotation actions, ruling out the possibility that subjects would be attending to intermediate poses. *Based on these observations, our model of mental rotation allows for discrete jumps in rotation space.*

**The Action setup provides strong evidence that rotation actions and match/mismatch decisions are guided by symbolic representations of objects.** Subjects performed an average of 1.05 rotation actions in the Action setup before reaching a decision. Even at high angular disparities, such as 180°, subjects performed on average fewer than two actions before reaching a decision (Fig. 6). This remarkable parsimony of actions suggest that humans dispose of powerful, abstract representations of objects to guide their rotation decisions. Moreover, these actions typically occur early in the trial (App. Fig. 9), indicating that they are indeed part of the process to come to a decision, and not simply a confirmation of an inner mental rotation process which would have already taken place. After the last rotation action, the final position of the rotated object is typically found to be distributed within the range [-45°, +45°] of the target object (Fig. 2.B). This imprecise alignment suggests that object representations used for comparison are not sensitive to this range of angles. The short decision time following the last action (App. Fig. 9.D), comparable to the overall response time in the case of 0° angular disparity, further confirms that no additional mental rotation process is taking place after the last action. These observations have led us to formulate a hypothesis on the symbolic nature of the representations used by human subjects to perform mental rotation tasks (described in the next section), which in turn guided the design of our model.

### 3.3. The Quadrant-Dependent Symbolic Representation Hypothesis

As a possible explanation for the parsimonious use of actions by human subjects and their tendency, in our VR experiment, to place objects at an angle between [-45°, +45°] relative to each other before making a similarity decision, we introduce the *Quadrant-Dependant Symbolic Representation Hypothesis* (hereafter referred to as the Quadrant Hypothesis). Under this hypothesis, candidates may mentally place objects into visual 'quadrants' (Fig. 3) (or frames of reference (Hinton & Parsons, 1981)), yielding a quadrant-specific symbolic encoding of objects, and thus effectively reducing the mental rotation task to switching between these discrete quadrants until the objects appear aligned. Specif-

ically, by dividing the 360° rotation space into four equal quadrants, each object can be assigned to one, allowing decisions on rotation actions and similarity judgment to be based on quadrant membership rather than exact angular position. Actions are consequently limited to rotating the shape by (i) one quadrant clockwise, (ii) one quadrant counterclockwise, or (iii) two quadrants, greatly limiting the typical number of actions needed to solve the task.

Algorithmically, Shepard-Metzler shapes are composed of ten cubes and comprise three elbows. A natural way to describe these shapes symbolically is through the description of the 9 transitions between cubes, starting from the nearest cube to the viewer, and where each transition corresponds to one of six possible directions: 1) U, up; 2) D, down; 3) B, back; 4) F, forward; 5) L, left; and 6) R, right. We adopt this sequential description as the symbolic representation of the object on which our quadrant hypothesis acts; the description remains the same within a quadrant, but changes with a quadrant switch, as shown in Fig. 3. A shape yields four unique symbolic descriptions (one for each quadrant) when starting from the nearest cube (indicated by the blue block in Fig. 3). A precise mathematical description of the Quadrant Hypothesis using group theory is given in App. D.

While humans may not use this exact coding scheme (see App. B), it provides a tractable abstraction which allows our model to capture key aspects of subjects' behavior in our experiments (see Sec. 5. Results). In our model, we assume that the rotation actions apply to spatial representations, consistently with elements of the literature (see Sec. 2), and that the decision strategy only, involved in similarity judgments and mental rotation choices, is made over the comparison of symbolic representations.

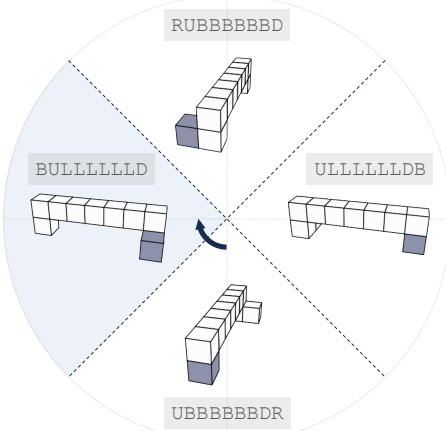

*Figure 3.* The Quadrant Hypothesis: a Shepard-Metzler shape has 4 possible descriptions depending on the reference frame (or viewpoint). Each description consists of the sequence of transition between blocks as observed from the given viewpoint. U: up, D: down, B: back, F: forward, L: left, R: right.

# 4. A Computational Model of Mental Rotation

We developed a computational model of human mental rotation. Our desiderata were that the model (1) be composed of neuron-like units organized into a network; (2) achieve at least human-level accuracy on the mental rotation task; and (3) be consistent with established accounts of human behavior in the literature and our experimental observations. In this section, we describe the components of our model and how they work together to solve the mental rotation task. For more details on the model and training see App. E.1.

## 4.1. Model Description

**Module I for Spatial Representation.** For the first module, we adopt the equivariant architecture proposed by (Dupont et al., 2020) (see Fig. 4 and caption for details), which is a convolutional auto-encoder trained on 2D images of centered 3D objects from varying viewpoints. During training, pairs of images of the same object at different poses are encoded, and each latent is rotated to the other's pose for reconstruction.[2] This enforces spatial structure in the latent space, resulting in a latent representation that is equivariant to the group of 3D rotations $SO(3)$. Note that this is achieved without 3D supervision, in line with human experience. At inference time, novel 2D views can be synthesized by applying a 3D rotation matrix directly to the latent representation. This architecture is used because it produces a **spatial representation**, thus enabling one of the key functional aspect of mental rotation: manipulation of internal representations that are analog (spatially structured) rather than symbolic, and formed by the 3D structures inferred from 2D views consistent with the literature cited in Section 2. The architectural backbone and core training procedure from (Dupont et al., 2020) are unchanged for the purposes of our model; the model was simply trained from scratch on our custom dataset of Shepard-Metzler objects. This module is trained on 50,000 image pairs (see App. E.2 for more details).

**Module II for Symbolic Representation.** The second module is a Vision Encoder–Decoder architecture named Vision Symbolic Model (VSM), which is composed of a Vision Transformer (ViT) encoder (Dosovitskiy et al., 2021) and an Autoregressive Transformer decoder (Vaswani et al., 2017). VSM is trained from scratch, end-to-end, to map the 3D latent representations of object views obtained from the first module's into their symbolic descriptions, as described in our Quadrant Hypothesis (see Section 3.3). We apply the same augmentation procedure as the equivariant neural renderer (EqRN), but this time to enforce equivariance at the symbolic level. We define the dataset for this second module

by extracting spatial latent representations using the frozen EqNR encoder from a set of 2D views of the same 3D objects used to train the EqNR, but with constrained poses: a fixed elevation angle ($\theta = 25°$) and varying azimuth angles ($\phi$), matching the viewing conditions of our VR experiments. The outputs are the symbolic sequences of the corresponding Shepard-Metzler objects (see Fig. 4). This module is trained on 201,600 images-symbolic descriptions pairs (see App. E.2 for more details).

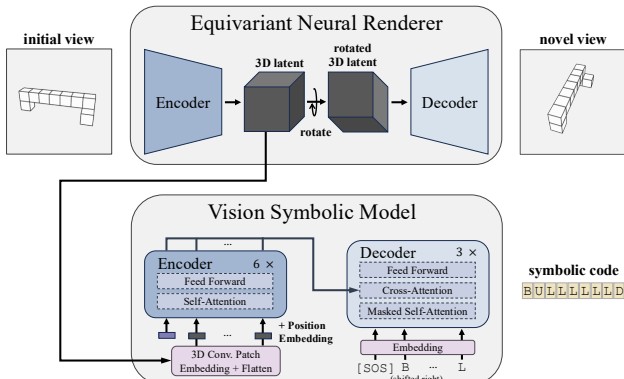

*Figure 4.* **Module I:** The encoder of the Equivariant Neural Renderer consists of a cascade of 2D residual convolutional blocks, an inverse 2D-to-3D projection, and a cascade of 3D residual convolutional blocks to produce a structured 3D latent representation of a view. The decoder is a transpose of the encoder that reconstructs the input image, or generates a novel viewpoint when a 3D rotation is applied on the latent space. The rotation operation consists of a smooth reindexing of units of the 3D-organized latent space through application of a rotation matrix to each unit's spatial indices, followed by an interpolation to remap the activities of units on the rotated grid. **Module II:** The Vision Symbolic Model consists of a Vision Transformer as the encoder and an Autoregressive Transformer as the decoder. It is trained end-to-end from scratch to map 3D latent representations into symbolic descriptions.

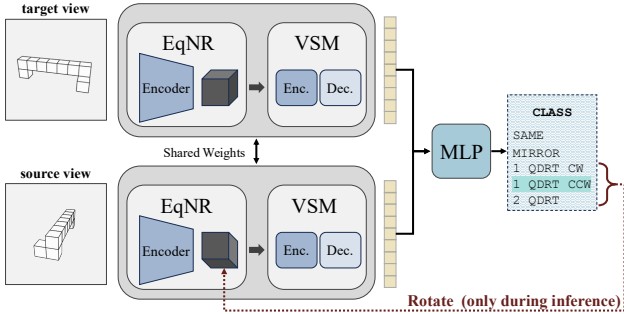

*Figure 5.* **Module III:** The Multi-Layer Perceptron (MLP) consists of three layers, each followed by a BatchNorm layer and a ReLU activation, except for the final layer, which uses a Softmax. It takes as input a concatenation of symbolic logits in a Siamese fashion, and outputs one of the five classes it is trained on: similar pair (SAME), mirror pair (MIRROR), one quadrant clockwise (1 QDRT CW), one quadrant counter-clockwise (1 QDRT CCW) or two quadrants apart (2 QDRT). Note that no rotation actions are applied during the training of this module.

---

[2]We do not explicitly propose a neural mechanism for this rotation operation, but see App. B for speculations on how it may be implemented in a biologically plausible neural system.

**Module III for Similarity Decision and Action Taking.**
We complete the architecture with the third and last module: a standard Multi-Layer Perceptron (MLP) composed of three hidden layers. The MLP operates on pairs of symbolic descriptions to make similarity judgment or rotation decision by identifying quadrant membership in accordance with the Quadrant Hypothesis. This third module works as follows (see Fig. 5): given a pair of images—the target and source views—the EqNR and VSM act as feature extractors applied independently to each input image; the first module extracts spatial representations and second module maps them into symbolic ones. The two resulting symbolic representations are concatenated in a Siamese-like fashion and fed into this third module, which assesses the implicit angular disparity between the two 2D views, outputting either a similarity decision (if views fall in the same quadrant) or their relative quadrant disposition, thus simplifying subsequent similarity judgments. This module is trained on 38,400 mental rotation tasks (see App. E.2 for more details).

### 4.2. Performing the Mental Rotation Task

To test our model on the mental rotation task, we created a dataset of held-out Shepard–Metzler objects that were not used during training of any of the three modules (thus avoiding trivial memorization effects), but were the same shapes employed in the VR experiments, though not displayed at the exact azimuthal angle. Each sample consists of a pair of views, termed target and source views, where each view is an image depicting either the same or a mirrored object, mimicking the stimulus framework of human mental rotation tasks. The dataset contains a balanced number of match (same) and mismatch (mirror) pairs, and within each similarity category, trials are balanced across angular disparities ($\Delta\phi = 0°, 60°, 120°$ or $180°$). The axis of rotation is fixed (Y-axis), with views sampled at varying azimuths ($\phi_{target} \in [-180°, 180°]$, $\phi_{source} = \phi_{target} + \Delta\phi$) but at fixed elevation ($\theta = 25°$), just like in the VR experiments.

Turning to the model architecture, it consists of three independently trained modules each specialized to handle a specific step in the processing pipeline (see Fig. 5): Module I extracts a 3D spatial representation from images, Module II maps these spatial representations into symbolic ones, Module III produces, from two symbolic representations, the similarity decision (match or mismatch) or the next rotation action to take (one quadrant clockwise → rotate $90°$, one quadrant counter-clockwise → rotate $-90°$, or two quadrants apart → rotate $180°$). When a rotation action is predicted, it is only applied to the 3D spatial representation of the source view (from Module I), which is then again reprocessed through Module II and Module III. This iterative mechanism enables the model to 'align' the symbolic representations of the source and target views.

Performance is measured as the accuracy in classifying trial pairs as match or mismatch. The number of rotations taken before the similarity decision (match or mismatch) by Module III provides a measure of the model's action count, analogous to thumbstick actions in the human experiments. The trial is stopped and counted as "failed" if, after six actions, the model does not predict a similarity decision.

## 5. Results – Performance and Comparison with Human Behavior

In this section, we report the model's performance and describe how it matches (and where it fails to) account for human behavior. Then, through systematic ablations, we show the necessity of each component of our model to perform the task and account for human behavior.

**The model performs well on the mental rotation task, with comparable accuracy to human subjects.** The overall accuracy is $96.13\%$; with $96.39\%$ for match trials and $95.87\%$ for mismatch trials, performing equally well on both types of pairs. These results indicate that just like humans ($91.14\%$ in No-Action and $95.33\%$ in Action setup, see App. Table 2), the model can successfully solve in-depth rotation tasks using only images.

**The model adequately captures the typical number of actions taken by humans for different angular disparities.** Looking at Fig. 6, for successful match trials across the different angular disparities, the model takes on average nearly as many actions as humans do in order to solve the task; dominated by 0 action for $0°$ disparity trials, and dominated by 1 action for the other disparities. This consistency supports the plausibility of the modeled mechanism to reflect the underlying cognitive processes involved in human mental rotation. Note that although humans' action counts are dominated by 1-action trials, they exhibit more variability in the number of actions within and across conditions (in particular at $120°$ and $180°$) than our model, which consistently performs 0 or 1 action. This variability could arise from noise inherent to human interaction with the experimental setup, including inconsistent thumbstick control, strategy switching, fatigue, and other sources of behavioral variability. In contrast, our model is deterministic and does not capture these types of fluctuations.

**Linear reaction times are partly explained by the model.** Our model can explain differences in reaction times between the $0°$ condition, the $60°$ condition and larger angles. Indeed, our model takes an average of 0.00 mental rotation actions at $0°$, 0.71 actions at $60°$, and 1.03 actions at $120°$ and $180°$. If we attribute a fixed duration to each action, we predict an approximately linear reaction time as a function of angle from $0°$ to $120°$. However our model cannot explain different reaction times at $120°$ and $180°$, as it almost al-

ways systematically predicts one rotation action in these two conditions. We ran additional analyses (see App. E.3): first we analyzed whether human reaction times correlated with the number of actions across trials at fixed angular disparity, but found no significant relationship; then we tried to model human data at a single-trial resolution, but inter-individual variability did not allow us to draw meaningful conclusions; finally, error consistency measures showed weak agreement among humans and none between the model and humans.

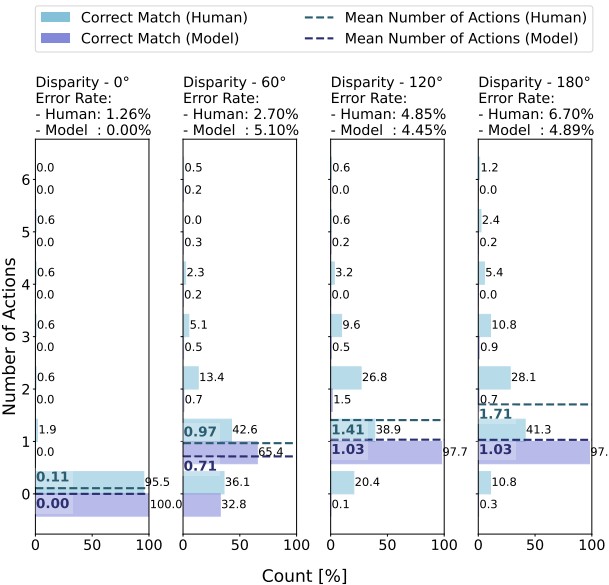

*Figure 6.* Distribution of number of actions taken per trial for Humans (Blue) and our Model (Purple) across angular disparities $(0°, 60°, 120°,$ and $180°)$. Each panel shows the normalized distribution of action count per successful match trials, along with the error rate and average number of actions taken for Humans and our Model.

## 5.1. Ablations

We performed extensive ablations to justify the role of each component in our model. A summary of the ablation study can be found in Table 1.

**We tested a classical Siamese architecture on the task.** Feed-forward architecture, such as ResNet or ViT (see App. Fig. 17), were used as Siamese encoders, each taking one object view as input (i.e., an image), and trained directly on the mental rotation task (same or different). While these models achieved near-perfect accuracy on training objects, they failed on novel test objects, except when object rotations were restricted to the image plane. This contrasts with human performance, which is robust to both in-plane and in-depth rotations (Shepard & Metzler, 1971). Additional details on this ablation are provided in App. E.4.1.

**We tried removing the equivariant encoder component of our model.** We directly fed the input images to the sym-

bolic encoder (VSM). We adapted the ViT patch embedding by replacing the 3D convolution with a 2D convolution, as the input is now a 2D image. The VSM was then trained on the task of predicting the sequential symbolic description of the object. Trained directly in pixel space, the symbolic encoder failed to reliably predict the symbolic description, such that the mental rotation task could not be solved.

**We tried removing the symbolic encoder component of our full model.** We conducted a three-stage ablation study to assess the necessity of symbolic module (VSM). For each stage, we only retrained the MLP. First, we replaced the entire VSM with a single frozen 3D convolutional layer (randomly initialized) that directly projects the EqRN latent representation to a 54-dimensional vector. Second, we only kept the frozen patch embedding from the ViT encoder, while in the third stage, we retained the entire ViT encoder and ablated the VSM's autoregressive decoder. For stages two and three, the spatial tokens were also projected into a 54-dimensional vector. Next, we tested the model on the mental rotation task for each ablation stage. We observe that the model can solve the mental rotation task by taking actions only when the ViT encoder is retained, achieving an accuracy of $90.71\%$, against $38.46\%$ for full VSM ablation and $36.14\%$ when only patch-embedding is kept. App. Fig. 16 shows the number of actions for the ablation where the ViT encoder is kept, note that we observe a performance drop for condition $60°$ and $120°$. In view of these results, one might wonder about the relevance of the autoregressive decoder. We adopted an autoregressive approach because it allows flexibility in the number of cubes forming the shapes (whereas a pure ViT only works on shapes with a fixed, predefined number of cubes).[3]

**We tested whether actions were needed at all for our architecture to solve the task.** We trained the MLP to directly and only predict whether the two symbolic descriptions (concatenated as input) corresponded to the same or different objects. We found that this approach was successful in performing the task ($97.03\%$ accuracy, comparable to the full unablated model). However, it was unable to account for human behavior, as it did not need to take *any* action. This ablation raises a fundamental question about mental rotation and imagery: why would humans need to take sequential mental (or manual) actions to solve the task, when a built-in invariant decoder trained on an adequate symbolic representation of the objects would suffice? (See speculation in App. B). We tried reducing the size of the MLP by going from a three-layer model to a one-layer model. When this one-layer model was trained to predict actions and similarity judgments, it solved the mental rotation task with an accuracy of $96.05\%$, while still recapitulating the typical number

---

[3]Note that the autoregressive approach exhibits a sequential encoding time consistent with eye-tracking studies, but lacks a saccadic mechanism to account for eye movements (see App. B).

*Table 1.* Summary of the Ablation Study.

| Model | Schematic | Accuracy | Number of Actions |
|---|---|---|---|
| Siamese Encoders (Baseline) |  | $\sim 50\%$ (chance level) on test objects | **N/A** (no actions taken) |
| Module I Ablation (w/o Equivariant Rep.) |  | **N/A** (the symbolic module fails to encode the object) | **N/A** (no actions taken) |
| Module II Ablation (w/o Symbolic Rep.) |  | $\sim 90\%$ (If ViT retained), $\sim 38\%$ (Otherwise) | Small numbers of actions taken, performances drop for some conditions. |
| Module III Ablation (w/o Actions) |  | $\sim 97\%$ on test objects | **N/A** (no actions taken) |
| Ours (Full Model) |  | $\sim 96\%$ on test objects | Small numbers of actions taken, compatible with humans. |

of actions. However, when trained to predict similarity judgments only, the accuracy dropped to 86.23%. This shows that, with sufficient hidden layers, a feedforward neural network trained only on similarity judgments can approximate rotations at the symbolic level or become invariant to them. Conversely, it demonstrates a slight computational advantage of the mental rotation approach (requiring fewer layers) compared to a purely feed-forward approach.

## 6. Discussion

**Novelty.** Here we propose a first neural model of human mental rotation, which (1) reproduces the human ability to compare Shepard-Metzler shapes across viewpoints, (2) captures subjects' sequence of actions in an interactive version of the task, and (3) partly explains subjects' reaction times. Our model adds to a recent collection of neural models of the human and primate ability to perform spatial reasoning tasks (Hamrick, 2019; Nayebi et al., 2023; Thompson et al., 2024; O'Connell et al., 2025; Khajuria et al., 2025; Mohammadi et al., 2025). In line with our own conclusions, these models use similar components to ours, such as equivariant encoders (or more general 'world models'), neuro-symbolic encoders, and agents performing simulations in latent spaces. Nevertheless, none of these previous models specifically address the problem of mental rotation. Closest to our work, O'Connell et al. (2025); Bonnen et al. (2026) propose models of 3D visual inference which they

use to estimate the similarity of objects seen from different viewpoints, a task related to mental rotation. Their models are in remarkable agreement with human judgments of similarity. Their models rely on a different set of components than ours, which does not allow to perform sequential actions; in (Bonnen et al., 2026), however, reaction time correlates with the layer of the model at which the solution is found. In another recent study, Leadholm et al. (2025) propose a novel cortically-inspired architecture able to perform symmetry transformations. Their work is difficult to directly compare to ours because it is conceptually very different and requires the sense of touch. On the machine learning side, models have been proposed to solve mental rotation tasks and more general visual inference tasks (Poggio & Edelman, 1990; Jaderberg et al., 2015; Tuggener et al., 2025; Kotar et al., 2025; Bardes et al., 2024; Keller & Welling, 2021; Wang et al., 2025; Mason et al., 2026), but none of them have attempted to account for human behavior with the level of scrutiny that we propose in this study.

**On Reaction Times.** While we have mainly focused on explaining *actions* taken by humans in our interactive version of the experiment, our model design has been guided by observations on reaction times, and it informs us on the mental processes these reaction times could correspond to. First, we found in our non-interactive version of the experiment a variability of response times at large angular disparities which puts in question the hypothesis of a fixed and con-

tinuous rotation speed at 60°/s, first posited by Shepard & Metzler (1971). We also observed a similarly large variability of response times in an open dataset produced by Ganis & Kievit (2015), confirming our own results. As further evidence for discrete jumps, in our interactive experiments subjects naturally performed ballistic movements with the joystick, and were unperturbed by having the objects disappear during the rotation. Both these observations guided our modeling effort towards discrete actions in rotation space. In turn, our model provides a plausible explanation for the linear response time curves. Indeed, as the angular disparity between Shepard–Metzler shapes increases, the chance that they end in different 'quadrants' and thus have distinct symbolic descriptions also rises, which in turn necessitates one or more sequential actions. However, our model cannot directly explain the increase in response time from 120° to 180°, as both these angular disparities necessitate at least one action, and typically takes no more than one action both in humans and in our model. One hypothesis for slower response times at 180° than 120° is that the neural implementation of rotation actions in the brain would take more time for large angles than small angles (see speculation paragraph 'Mental Actions: Why and How?' in App. B).

**Symbolic or Spatial representations? We Answer 'Both!'** Another central question in modeling mental rotation pertains to the nature of the representations on which the different parts of the process operate (e.g., encoding visual input into internal representations, transforming them, assessing their similarity) (Pylyshyn, 2002; Kosslyn et al., 2006; Brogaard & Gatzia, 2017): are they carried out in a symbolic or a spatial domain? The linear relationship between reaction time and angular disparity found by (Shepard & Metzler, 1971) has been taken as evidence that the entire process operates on a 3D-structured spatial representation. On the other hand, our VR experiments reveal that humans rely on a remarkably low number of actions to align the shapes. Furthermore, subjects do not seek perfect alignment between shapes before taking a decision, but rather care about placing the shapes in an interval of [-45°,45°] with respect to each other. These behavioral signatures suggest that, while rotations might be applied to a spatial representation of the object, decisions such as similarity judgments or rotation choices are made from a more abstract, symbolic representation of the object. Our modeling effort confirms the feasibility of using neurally-obtained symbolic descriptions to guide similarity judgments and rotation actions on spatial representations, and our model provides a good match to human behavior. We note that in our model the rotation actions could also have been directly applied to the symbolic representations, and we do not know whether the 3D latent space is fully needed. However, previous literature does seem to indicate that analog spatial representations are indeed used, which comforts us in our hybrid choice (Cooper, 1976; Cooper & Shepard, 1978; Zacks, 2008, p. 124). Interestingly, in another task of spatial simulation, Sosa et al. (2025) find that a hybrid symbolic-spatial strategy is required to explain mental simulation capabilities of primates. It is a tantalizing generalization to think that our mind is able to carry out mental actions at different levels of abstraction, depending on the visuo-spatial task at hand.

**Relevance for Machine Learning.** Here our main objective was to better understand how humans can perform mental rotation. Following the maxim attributed to Feynman, *'What I cannot create, I do not understand'*, we sought to build a viable neural implementation of mental rotation by assembling existent components of modern deep learning, this work could in turn inspire new machine learning solutions for spatial reasoning tasks. It has indeed been shown that current deep learning approaches struggle on spatial reasoning tasks (e.g., Yang et al., 2025; Kosoy et al., 2025), and in particular on mental rotation tasks (Stogiannidis et al., 2025) and on the recognition of known objects rotated from their upright pose (Alcorn et al., 2019; Abbas & Deny, 2023; Ollikka et al., 2025; Cooper et al., 2025).[4] As for video-language models, although they perform surprisingly well on vision tasks they were not trained on (Wiedemer et al., 2025), object rotation is a failure case. Moreover, recent theoretical work by Perin & Deny (2025) has shown that traditional architectures are ill-suited to learn and generalize symmetry-invariant representations from data (e.g., invariance to pose). Our work suggests that explicitly enforcing the right representational inductive biases, rather than hoping they emerge from scale, may be a direction for closing this gap. The closest family of machine learning models to ours is the family of *latent equivariant operator methods* (Dinh & Deny, 2026), and notably JEPA (LeCun, 2022). Like JEPA, our model simulates inferred actions in a latent space between two latent views. Our model differs from JEPA in that (1) it enforces equivariance to rotations at an intermediate stage of the model, producing structured, spatially isomorphic representations; (2) actions are taken by a neural agent operating from neuro-symbolic representations and onto these spatial representations, in a sequential manner and *via* a recurrent pathway. It would be interesting to test the ability of existing JEPA architectures to perform mental rotation tasks. Conversely, it would be interesting to adapt and apply our model to different and more general spatial reasoning tasks, and compare the performance with existing methods. We leave this for future work.

**Supplementary Discussion.** For additional discussion on the necessity of action taking, or biological plausibility, data diet and coding scheme, see App. B.

---

[4]Note that it is debated whether recognition of everyday objects engages the same cognitive processes as mental rotation (Searle & Hamm, 2017).

## Acknowledgements

Credit Statement: R.K. developed the model of mental rotation, participated to the experimental design and to the data analysis. D.F. conducted the experiments, participated to the experimental design and to the data analysis. A.K. created the dataset and produced some of the ablations. Q.L. developed the VR application. S.D. conceptualized the study and advised on all parts of the work. All authors participated to brainstorming meetings. R.K. and S.D. mainly wrote the paper.

We thank Matias Koponen, Petteri Kaski, Kaie Kubjas, Sanni Kilpeläinen, Riina Pöllänen, Carlos Sevilla Salcedo, Aleksandr Krylov and Luigi Acerbi for related investigations on mental rotation not reported in this study, and Victor Boutin, Tyler Bonnen, Etienne Thuillier, Otso Haavisto, Robin Welsch and the BRAIN lab for useful discussions. We are also grateful to the Aalto Science-IT project for providing us with the computational resources.

Funding Source: Research Council of Finland grant to S.D. under the Project "Neuroscience-inspired Deep Learning": 3357590.

## Impact Statement

This paper presents work whose goal is to advance the fields of Machine Learning and Cognitive Science. There are many potential societal consequences of our work, none which we feel must be specifically highlighted here.

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

# Appendix

## Contents

All code and experimental data used in this work are available at github.com/rkhz/menrot.

## A. Additional Literature on Mental Rotation

**Mental and Manual Rotation.** Another question pertains to whether mental rotation and manual rotation—where subjects can actually manipulate objects to compare to align them—share common mental processes. Neuroimaging studies (Zacks, 2008) suggest that motor areas are involved in some mental rotation tasks, indicating colocation in the brain of some of the cognitive processes involved in manual and mental rotation. More directly, Chu & Kita (2011) show that subtle hand movements can help subjects perform mental rotation tasks, especially for subjects with little training. Moreover, experiments in a manual rotation setup with Shepard-Metzler shapes showed that manual and mental rotation exhibited similar chronometric curves (Wohlschläger & Wohlschläger, 1998). Furthermore, when asked to perform hand movements that did not reorient the stimuli during the mental rotation tasks, concordant rotational directions facilitated, whereas discordant directions inhibited, mental rotation (Wohlschläger & Wohlschläger, 1998; Wohlschläger, 2001). Taken together, these observations strongly suggest that mental and manual rotation share at least some common processes, both algorithmically and physically in the brain. *Building on this hypothesis, we used manual rotation experiments as a window into the cognitive process of mental rotation and to further constrain our model of mental rotation. Our model is equally capable of accounting for mental and manual rotation experimental findings.*

**VR Environment vs. 2D Images.** Studies (e.g., Stark et al., 2024) have shown that essential results on mental rotation (e.g., linear response times with angular disparity) can be reproduced in a VR environment. Interestingly, Stark et al. (2024) show that subjects perform mental rotation slightly faster in a VR environment than from viewing 2D images. This could be due to the perceptual benefits of parallax, or simply to the additional concentration afforded by being in an immersive environment. *We decided to run our experiments in a VR environment, as it is a more ecologically relevant setup than 2D images.*

## B. Additional Discussion

**Mental Actions: Why and How? Speculations on the Role and Biological Implementation of Rotation Operations.**
Why are humans taking actions (mental or manual) at all, when our model does not strictly need to? Indeed we found that a MLP with enough hidden layers, trained to directly predict match/mismatch from the symbolic descriptions of objects, did as well on the task as our full model equipped with an agentic MLP. We believe that this is a fundamental question that here we can only speculate on. One hypothesis is that the ability to carry out mental actions allows humans not only to detect whether two shapes are the same, but also to know what angle of rotation exists between them, an information that would be lost by a rotation-invariant shape detector (an argument also made by Mohammadi et al., 2025). Another question pertains to *how* rotation actions could be implemented in a neural system within the brain. We can speculate that gated lateral recurrent connections in cortex could achieve such reindexing of neurons: when the gates are activated by a decision-making brain region (analog to our agent), the lateral connections take effect and progressively reindex neurons at different spatial positions (corresponding to a rotation of the latent representation), in the spirit of *delay lines* and *shift register networks* (White et al., 2004). The rotation actions could be controlled by a decision-making region such as the prefrontal cortex (PFC) and communicated *via* a feedback pathway to visuo-spatial regions, such as the primary visual cortex (V1). Interestingly, such mechanism for rotation predicts the duration of the rotation operation to be proportional to the rotation angle, which could explain why subjects are slower in the 180° condition than in the 120° condition in the No-Action setup, but not in the Action setup, where actions are carried out manually and at very high speeds.

**Biological Plausibility of Backpropagation, Data Diet and Coding Scheme.** First, all the components of our model rely on backpropagation, which is largely deemed biologically implausible because of the weight transport problem (although see Lillicrap et al. 2020). It would be interesting to investigate neural models trained with local learning rules only (Illing et al., 2021; Siddiqui et al., 2024; Parthasarathy et al., 2025). Second, our model needs training on a very specific data diet of Shepard-Metzler shapes, seen in large numbers, and the trained model is unable to generalize to shapes outside of this precise family. In contrast, it is unlikely that humans would need to be exposed to that many shapes from the same family before being able to generalize to them (Ayzenberg et al., 2025), which suggests that more data-efficient learning mechanisms exist in the brain. One hypothesis is that a compositional description of objects may be achieved in the brain, allowing to encode unfamiliar objects from known primitives. Relatedly, Just & Carpenter (1976), by tracking gaze patterns, have shown that saccades focus sequentially on different elbows of the Shepard-Metzler shapes, suggesting that humans encode Shepard-Metzler shapes piece by piece. Their study offers an alternative—albeit more sophisticated—coding scheme compatible with our Quadrant Hypothesis (see Section 3.3). It has also been suggested by Fu et al. (2025) that saccades could play a key role in mental rotation, by being integrated in regions downstream from the visual system (Bonnen et al., 2022). It would be interesting to equip our model with a saccadic mechanism (like in Cheung et al., 2017; Thompson et al., 2024; Wu & Xie, 2024), which could participate in building compositional descriptions of objects and perhaps allow for more data-efficient training. For other objects with different symmetries, we can speculate that other symbolic encoding schemes may apply (see Takano (1989) for an extensive and insightful discussion). Lastly, note that we adopted an autoregressive transformer for the sequential symbolic encoding of the directions of the cubes that form the Shepard-Metzler shapes (see Module II in Section 4.1). Thereby, the autoregressive approach takes an encoding time proportional to the length of the shape, which is compatible with the observed dependency of human reaction times on the number of cubes forming Shepard-Metzler objects (compare Shepard & Metzler (1971) and Shepard & Metzler (1988)), and with the sequential encoding of objects observed in eye-tracking studies (Just & Carpenter, 1976).

## C. VR Experiments

We conducted experiments in VR using a custom-made app built with Unity and powered by a Meta Quest 2 VR headset.

### C.1. Experimental Design

**Experimental Design.** During the experiments, the subject sits on a chair while equipped with a Meta Quest 2 VR headset. A pair of random 3D Shepard-Metzler shapes, which are composed 10 adjacent cubes arranged in a structure comprising three elbows, appear in front of the subject. The shapes are presented at 25° elevation, upright as if they were sitting on the floor, with varying azimuthal orientations around the Y-axis, corresponding to in-depth rotation; within each pair, objects differed by discrete relative azimuthal rotations of 0°, 60°, 120°, or 180°. After the pair is presented, the subject must answer as soon as possible whether the two objects are the same or a mirrored version of each other. The time the subject takes to press the button is measured and corresponds to the reaction time. The task sessions are presented in periods of

10–15 minutes, with breaks between periods. The app presents two different operating modes. For each mode, subjects experience 2 sessions, with each session containing 50 trials: 25 match trials and 25 mismatch trials. In the first mode (No-Action setup), the user cannot manipulate the 3D object orientation to bring both shapes into congruence, whereas in the second mode (Action setup), the user can manipulate the object with the help of a joystick. The object rotates along a fixed axis (Y axis). During the rotation, the object disappears from view, and only an annulus around the object remains visible, with a small ball traveling along this annulus to indicate the angle of rotation. Their actions on the joystick are measured and timed. In both setups, participants are instructed to solve the task as fast as possible. In the action setup, participants are instructed that they may use the joystick to help them, but that it is not necessary to align the shapes before answering the mental rotation task.

**The No-Action Setup.** In this configuration (App. Fig. 7), the participants must mentally rotate one of the shapes to determine, by pressing decision buttons, whether they are the same shape in different orientations (match) or mirror versions of each other (mismatch). This setup mimics the original Shepard and Metzler experiment (Shepard & Metzler, 1971) but ported in a VR environment. The performance and reaction time are measured.

**The Action Setup.** This setup is similar to the no-action setup (App. Fig. 8), but now participants can manipulate the object using the VR thumbstick to rotate it in real-time, integrating real physical control into the mental rotation task. Introducing an 'action' component in addition to the reaction time. Interacting with the object causes it to disappear for the duration of the rotation, preventing participants from seeing intermediate poses; there is a green ball on a green circle indicating its position.

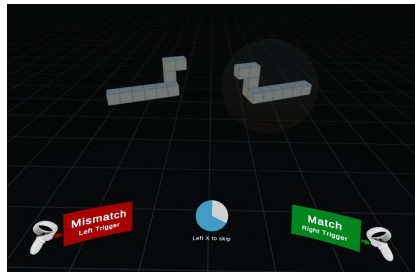
*Figure 7.* GUI screenshot for the No-Action Setup

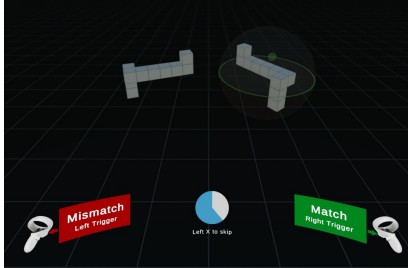
*Figure 8.* GUI screenshot for the Action Setup

### C.2. Participants

**Participants.** Adapted from Fernandes (2024): the study participants are volunteers, both male and female, they must have normal vision or vision corrected with contact lenses (no glasses), should be older than 18 years old and cannot have epilepsy, migraines, or be claustrophobic. The study counted 19 participants, 5F and 14M, aged range 22 to 30, mostly with a study background in engineering. In accordance with TENK national guidelines (the Finnish National Board on Research Integrity), this study required ethics approval with the collection of sensitive personal data; the data will be fully anonymized upon publication. The study was approved by an internal Ethics committee (IRB n°D/635/03.04/2024) and followed international guidelines from the Helsinki Convention (Association et al., 2013) as well as the TENK national guidelines. Participants were compensated 15€ for 2 sessions over two days (each session approximately an hour). We collected informed consent from each participant before the study in accordance with the Declaration of Helsinki guidelines.

**Instructions given to participants.** Before the session, the participants were instructed with the following text (Fernandes, 2024): *"This study examines the ability to mentally rotate and match two different 3D representations of the same 3D object portrayed from different orientations. You will use a custom-made VR app throughout the study. Random 3D shapes will appear in front of you representing the stimuli. After the stimuli is presented, you must answer as soon as possible if the two objects are the same or a mirrored version. For that you will press a button using your index fingers. Before the study starts, you will have the opportunity to test the app and get familiar with the interface. You will have 2 different sessions, one for each mode. You may take a break between sessions. During the first session, the shapes will be presented to you and you must simply answer if there is a match or mismatch between the two objects. During the second session, you are again presented with two shapes; however, you now have the opportunity to rotate one of the shapes as an aid mechanism to help you solve the problem. The goal is not to align both the shapes to verify the occurrence of a match but rather use this dynamic as a helping tool to help you mentally rotate the object. Use the joystick as intuitively as possible. You should be as focused as possible throughout the whole duration of the study."*

**Exclusion of 4 participants from the analyses based on atypical behavior.** We calculated the average reaction time for each subject's successful match trials with an angular disparity of $0°$. Using the interquartile range (IQR) method, we removed subjects whose average times fall outside the interval $[Q1 - 1.5 \times IQR; Q3 + 1.5 \times IQR]$, where Q1 and Q3 are the 25th and 75th percentiles of the average reaction times distribution, and $IQR = Q3 - Q1$. Four subjects were removed from the analysis because their average reaction time exceeded the upper bound $Q3 + 1.5 \times IQR$. The rationale behind this filtering is that extreme response times on average when viewing two identical objects at the same orientation, likely indicate that the subject was distracted, did not understand the task, or did not engage properly with the experiment.

**Trial coupling across participants.** Sessions are paired such that session $2n - 1$ display the same trials as session $2n$ (e.g., session 1 = session 2, session 3 = session 4, etc.). Each subject $k$ experienced two sessions for each setup: session $2k$ and $2k + 1$. This meant that the subjects shared a session with the previous subject and the consecutive one. This design enables the formation of two cohorts that are each exposed to the same trials: Cohort I composed of all odd sessions and Cohort II composed of all even sessions. Note that due to an experimental error in No-Action Setup, 6 subjects were instead presented two paired corresponding sessions (i.e., both sessions $2n - 1$ and $2n$). Although this prevented the formation of separate cohorts, we did not observe significant alteration in reaction times for these subjects.

## C.3. Measurements

*Table 2.* Average response times (RTs) and accuracy in the Action and No-Action setups.

| (Task) / Setup | No-Action | Action |
|---|---|---|
| Accuracy | 91.14% | 95.33% |
| Accuracy (Match) | 91.33% | 96.13% |
| Accuracy (Mismatch) | 91.50% | 94.53% |
| Mean RTs (Match) | 3.72 s | 3.40 s |
| Mean RTs (Mismatch) | 4.58 s | 4.21 s |

**Decomposing response time in the Action Setup.** The response time of the Action Setup can be dissected into three time intervals: (i) $T_p$, planning time; (ii) $T_a$, action time; and (iii) $T_d$, decision time. It is important to note that we cannot access this interval dichotomy if subjects did not utilize the joystick during the trials but instead simply pressed the decision button; then only their overall response time can be measured, just as in the No-Action setup. App. Fig. 9 reports the average of the three times intervals—$T_p$, $T_a$, and $T_d$—from successful match trials given joystick interactions, and across angular disparities of $60°$, $120°$, and $180°$. For trials with $0°$ disparity, we only report the overall response time from the Action Setup, excluding cases with joystick interactions; there are relatively rare (e.g., only 7 over 157 in the entire study), and are interpreted as accidental, since no action should be required in this orientation condition:

- **For $T_p$:** App. Fig. 9 shows that this time occupies approximately half of the overall response within the trial, that it is almost constant across all angle disparities ($\sim 1.75$ seconds), and it accounts for almost the overall response time for cases where objects are aligned; this is reminiscent of the common planning process observed by (Wohlschläger & Wohlschläger, 1998; Wohlschläger, 2001). This also indicates that, after this period and before any physical interaction with the object, the subject already has a good guess of the transformation needed to align the objects—typically selecting the shortest path to do the rotation.

- **For $T_a$:** Focusing on the joystick interaction time interval, we observe that it increase linearly with angular disparity increases, and it is the only that varies across angle differences; this is reminiscent of the linear trend observed by the seminal work by (Shepard & Metzler, 1971), a hallmark of the mental rotation. Note that during interaction, people can have intermediate small pauses, and those pauses are counted in the average of this interval.

- **For $T_d$:** Finally, we observe that this time period is constant across all angular disparities ($\sim 1$ second), and that it accounts for about half of the overall response time for cases where objects are aligned; we conclude that during this time interval there is no further mental rotation process taking place, and that this decision time is solely associated with reaction time to assess similarity and then press the corresponding decision button.

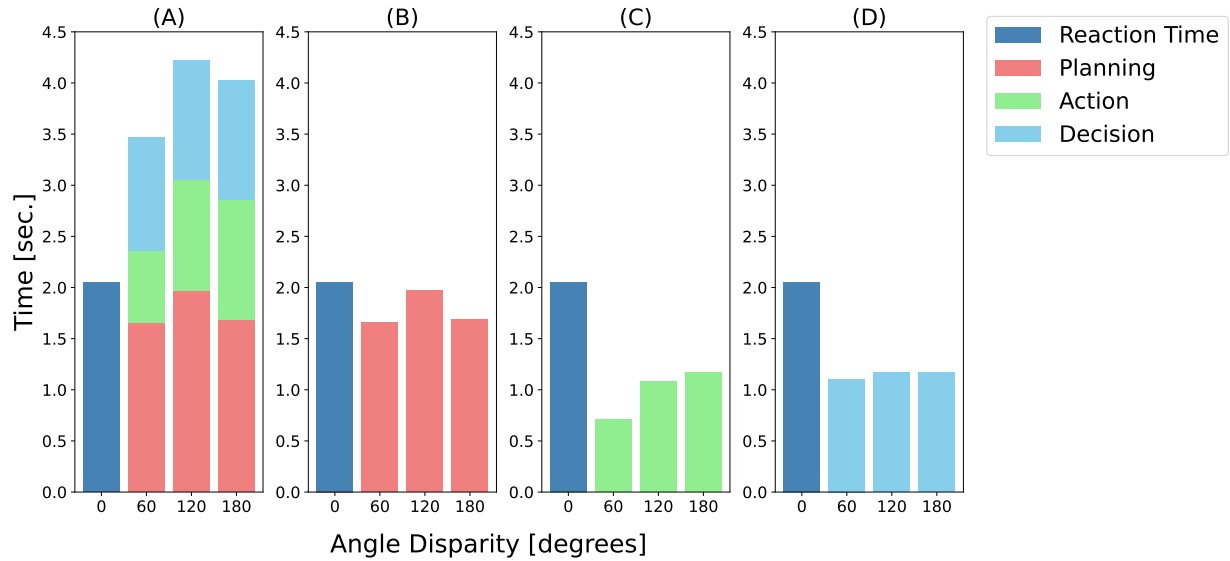

*Figure 9.* (A) Average reaction times in the Action setup, divided into three intervals: planning time ('Planning'); action time ('Action'); and decision time after last rotation action ('Decision'). For the 0° angular disparity condition we only report average total response time ('Reaction Time'), as typically no action was taken. In the other conditions, we only took into account trials where at least one action was taken (as the breakdown in three phases is otherwise impossible). (B-D) Present the phases in isolation for ease of comparison between angular disparity conditions.

**Counting the number of actions.** The raw angular position $\phi$ obtained from the interaction with the VR's thumbstick is first resampled at $f_s = 100$Hz. This sampling frequency is chosen so that it complies with the Nyquist theorem by ensuring $f_s > 2f_{max}$, where $f_{max}$ is the highest frequency present in the data. The angular velocity $\omega$ is then calculated from the resampled angular position values. The number of actions is determined by counting both upward and downward crossings of the angular velocity at a threshold of $150°$/s, where each pair of consecutive crossings marks the start and end of significant rotational movements; delineating distinct "actions" within the Action Setup (see App. Fig. 10).

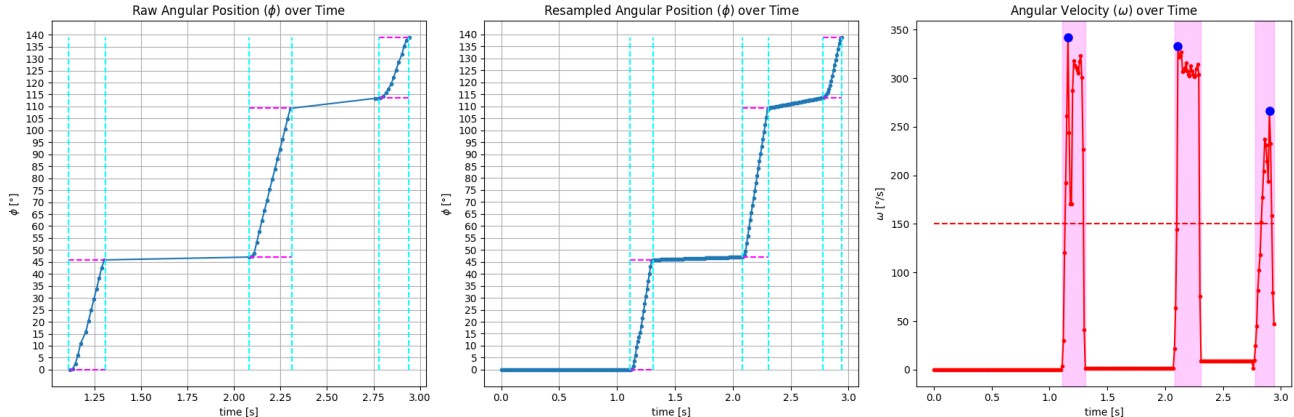

*Figure 10.* Counting the number of actions in a trial. (Left) Raw angular position of the movable shape over time. (Middle) Resampled angular position. (Right) Angular velocity of the shape over time.

## C.4. Additional Analyses

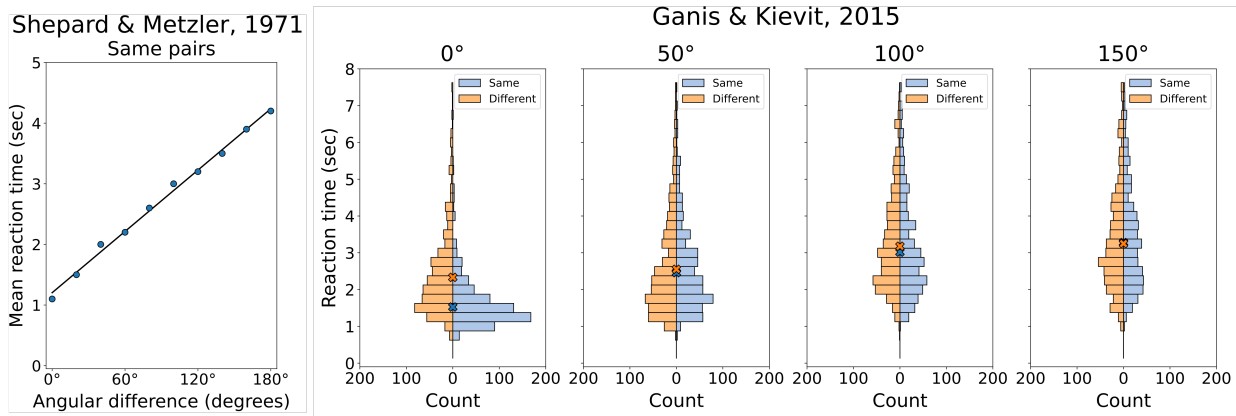

*Figure 11. Left:* Reproduction of the linear reaction times reported by Shepard & Metzler (1971). *Right:* Distribution of reaction times from the open dataset of Ganis & Kievit (2015). Like in our own experiments, a large variability can be found in reaction times for large angular disparities. At 150°, some responses can be very fast, almost as fast as the fastest responses at 50°, putting in question the constant rotation speed hypothesis. A similar amount of variability in response times could be found in single subjects (not shown), discarding the possibility that this variability could simply reflect different reaction time slopes across individuals.

# D. The Quadrant Hypothesis: Mathematical Formalism

The symbolic sequence is invariant under continuous rotations within each quadrant, but transforms equivariantly under the cyclic group $C_4$, corresponding to discrete 90° rotations around the vertical axis.

In other words, let the rotation angle of the object be $\phi \in [-\pi, \pi]$. We partition the circle into four quadrants:

$$Q_1 =] - \pi/4, \pi/4], \quad Q_2 =]\pi/4, 3\pi/4], \quad Q_3 =]3\pi/4, 5\pi/4], \quad Q_4 =]5\pi/4, 7\pi/4].$$

Within each quadrant $Q_i$, the symbolic encoding of an object $S(\phi)$ is invariant. Rotations by $k\pi/2$, $k \in \{0, 1, 2, 3\}$, map the encoding to another quadrant according to the group action $g$ of $C_4$:

$$\begin{cases} S(\phi_1) = S(\phi_2), & \text{if } \phi_1, \phi_2 \in Q_i \text{ where } i \in \{1, 2, 3, 4\} \quad \text{(invariance within quadrant)} \\ S(\phi + k\pi/2) = g_k \cdot S(\phi), & k \in \{0, 1, 2, 3\}, g \in C_4 \quad \text{(equivariance across quadrants)} \end{cases}$$

where $g^0 = e$ is the identity (leaving the sequence unchanged), $g$ permutes the symbolic letters as:

$$B \mapsto R, \quad R \mapsto F, \quad F \mapsto L, \quad L \mapsto B, \quad U \mapsto U, \quad D \mapsto D,$$

and $g_k = g^k$ for $k = 1, 2, 3$. Note that $U$ and $D$ do not change.

# E. Details of the Models

## E.1. Details of Architecture and Training

**Hyperparameters of Module I.** The details of the EqNR can be found in the appendix of (Dupont et al., 2020). The EqNR was trained for 200 epochs (early stopping at 187), on $W = 8$ Nvidia V100 GPUs using Distributed Data Parallel (DDP) with a batch size of 16 per GPU, using Adam with a learning rate of 2e-4 (scaled by $\sqrt{W/4}$), and the MSE loss.

**Hyperparameters of Module II.** The VSM consists of a ViT encoder and an autoregressive Transformers decoder. The ViT encoder includes a patch embedding implemented as a 3D convolution (input channels 64, output channels 384, kernel size 8, stride 8) followed by flattening, and six transformer encoder layers, each comprising layer normalization, multi-head attention layer (6 heads, dimension 64, embedding dimension 384) with a residual connection, followed by layer normalization and a feed-forward network with input dimension 384, hidden dimension 1536, and output dimension 384, using GELU activations and a residual connection. The Autoregressive Transformers decoder includes a simple embedding

with 7 tokens (start of sequence tokens and six possible directions of the symbolic description) with dimension embedding of size 384, followed by three standard transformers decoder layers, each comprising: a multi-head self-attention layer (6 heads, dimension 64, embedding dimension 384) with a residual connection and a layer normalization, a multi-head cross-attention layer (6 heads, dimension 64, embedding dimension 384) with a residual connection and a layer normalization, and finally a feed-forward network with input dimension 384, hidden dimension 2048, and output dimension 384, using ReLU activations, a residual connection and a layer normalization. The VSM was trained for 250 epochs (early stopping at 201), on $W = 8$ Nvidia V100 GPUs using DDP with a batch size of 32 per GPU, using AdamW with a weight decay of 1e-2, a learning rate of 2e-4 for the ViT encoder and 1e-4 for the Autoregressive decoder (scaled each by $\sqrt{W}$), a cosine annealing scheduler with a step per epoch of $\frac{\text{size of the training dataset}}{W \times \text{batch size per GPU}}$, and the Cross-Entropy loss with label smoothing of 1e-1.

Note that he latent space outputted from the EqRN encoder has shape $(64, 32, 32, 32)$, much larger than conventional ViT image inputs. To convert this into ViT-compatible tokens, and mostly for efficiency purposes, we apply a learnable 3D convolutional patch embedding with input channels 64, output channels $D = 384$ (the embedding size), and kernel size and stride 8, mimicking the standard ViT patchification. This produces a tensor of shape $(D, 4, 4, 4)$, which is rearranged into 64 tokens of size $D$. A learnable class token is concatenated to these tokens, resulting in 65 tokens. Learnable positional embeddings are then added, and the tokens are finally fed into the stack of $N$ identical vanilla Transformer encoder layers composing the ViT. The ViT encoder outputs 65 tokens of size $D$ that are passed as memory context $\mathbf{m} \in \mathbb{R}^{65 \times D}$ via the cross-attention layer of the autoregressive Transformer decoder. During training, we employ teacher forcing with shifted-input (Vaswani et al., 2017) to produce the symbolic sequence. The decoder receives as input $\mathbf{x}_{\text{dec}} = [[\text{SOS}], y_1, \ldots, y_{T-1}]$, a shifted version of the target sequence $\mathbf{y} = [y_1, \ldots, y_T]$ that it is trained to predict, where each token $y_t \in \{\text{U}, \text{D}, \text{B}, \text{F}, \text{L}, \text{R}\}$ and $T = 9$. Within the decoder's self-attention layers, causal masking ensures that each position $t$ can only attend to tokens at positions $\leq t$ (Vaswani et al., 2017). Finally, the training objective is the cross-entropy loss:

$$\mathcal{L} = -\sum_{t=1}^{T} \log P(y_t \mid \mathbf{x}_{\text{dec}}[1:t], \mathbf{m}), \tag{1}$$

so that each $y_t$ is predicted conditioned on past tokens and the ViT output. Note that for more clarity we omitted the batch dimension in the equations.

During inference, given an image $\mathbf{I}$ we extract its spatial presentation with the encoder of the first module $\mathbf{z} = \text{EqNR}_{\text{enc}}(\mathbf{I})$, which we then process using the encoder of this second module to produce the context $\mathbf{m} = \text{VSM}_{\text{enc}}(\mathbf{z})$. Finally, we use the autoregressive decoder of this second module to generate the symbolic sequence $\hat{\mathbf{y}} = [\hat{y}_1, \ldots, \hat{y}_9]$, by initializing the generation with the $\langle SOS \rangle$ token and iterating $T = 9$ times while attending to $\mathbf{m}$:

$$\begin{aligned}
\hat{y}_t &= \text{argmax}(\text{VSM}_{\text{dec}}(\mathbf{x}_{\text{dec};t-1}, \mathbf{m})), \\
\mathbf{x}_{\text{dec};0} &= [\langle SOS \rangle], \\
\mathbf{x}_{\text{dec};t} &= [\langle SOS \rangle, \hat{y}_1, \ldots, \hat{y}_t], \textbf{ where } t = 1, \ldots, 9
\end{aligned} \tag{2}$$

**Hyperparameters of Module III.** The MLP is composed of three-layer, the first layer has 256 to 64 units, the second layer has 64 to 16 units, each hidden layer followed by 1D batch normalization and ReLU, execpt for the final layer that uses a softmax activation. The MLP was trained for 1000 epochs (early stopping at 742), on $W = 1$ Nvidia V100 GPUs with a batch size of 64, using AdamW with a weight decay of 5e-2, a learning rate of 1e-3, and the Cross-Entropy loss.

The MLP is trained in a supervised manner on five classes. If the two objects fall in the same quadrant, it predicts whether they are 'same' or 'mirror' pairs. Otherwise it predicts the relative quadrant difference between the objects ('one quadrant clockwise', 'one quadrant counterclockwise', or 'two quadrants apart'). Note that no rotation actions are applied during the training of this final module. That being said, at inference time, if no similarity decision is made, the predicted quadrant disposition is used as a rotation action to be applied on the spatial representation: i) one quadrant clockwise $\rightarrow$ rotation $90°$, ii) one quadrant counter-clockwise $\rightarrow$ rotation $-90°$, two quadrants apart $\rightarrow$ rotation $180°$.[5]

---

[5]In the brain, such actions on spatial representations could be implemented via a feedback pathway from decision-making regions (e.g., prefrontal cortex) to visuo-spatial regions (e.g., V1), acting on a gated recurrent circuit within the visuo-spatial region to perform the rotation operation (see App. B. Additional Discussion).

### E.2. Datasets Descriptions and Splits

Shepard-Metzler objects are rendered using a custom rendering engine[6], based on *matplotlib*. Our custom rendering engine applies the classical rules of perspective projection. Self-occlusions are treated via the painter's algorithm. No shades were applied. Objects are centered within a sphere, and rendering a viewpoint requires the object's symbolic description along with camera distance, azimuth, and elevation. The renderer engine uses a coordinate system where the Z-axis points outward from the screen toward the viewer, the Y-axis points downward from top to bottom, and the X-axis points rightward from left to right. In this work, each viewpoint is a 2D image of size $(3, 128, 128)$ with a white background, captured from a fixed camera distance of 25, with all cube faces of the object in white and edges in black.

Shepard-Metzler objects are composed of 10 cubes and comprise three elbows. There are 288 unique Shepard-Metzler objects, from which 200 are used for the training set, 30 for the validation set, and 58 for the testing set. Each object has 6 fundamental shape-views, resulting in 1200 fundamental shape-views for the training set, 180 for the validation set and 348 for the testing set. For each shape-view, 4 distinct symbolic descriptions are obtained by rotating the shape around its vertical axis, resulting in a total of 4800 symbolic descriptions in the training set, 720 in the validation set, and 1392 in the testing set. See App. Fig. 12 for more details on the shape-views and corresponding symbolic-descriptions.

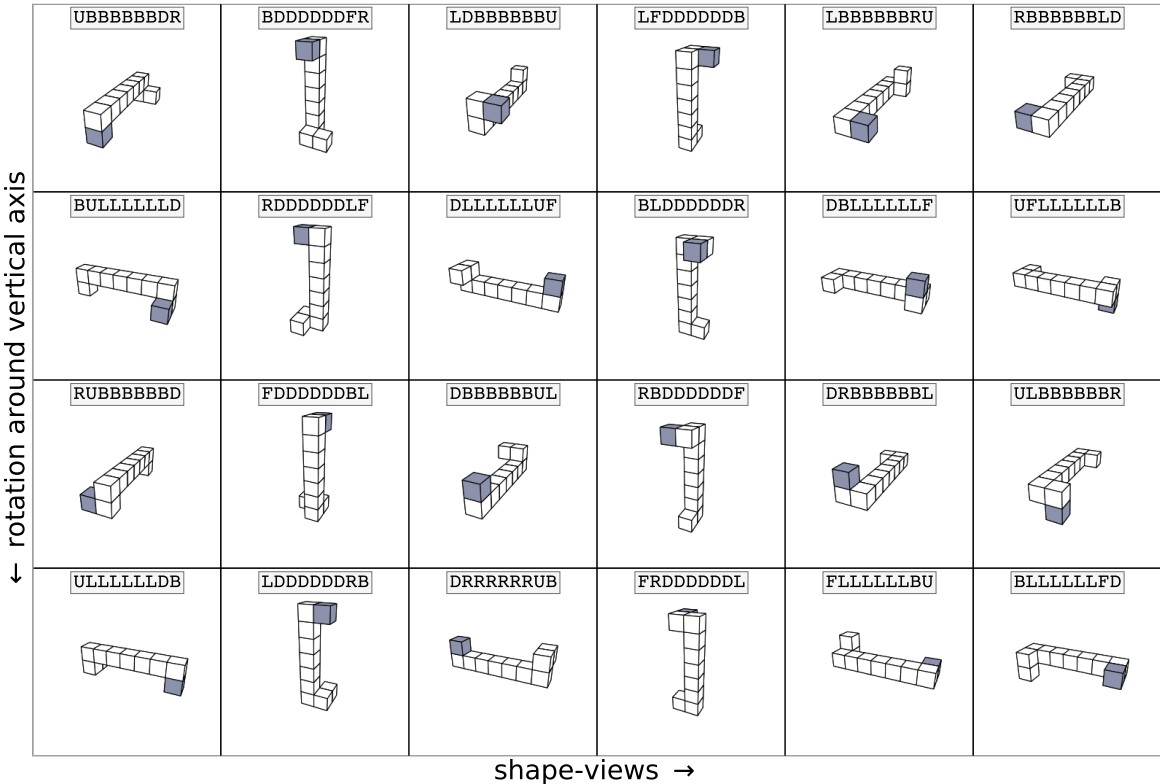

*Figure 12.* Set of all unique symbolic descriptions we can get from a single Shepard-Metzler object. To define the 6 shape-views of a Shepard-Metzler object, we first enclose the object in a larger imaginary cube whose faces are colored: top–bottom (Red–Yellow), front–back (Blue–Green), and left–right (Cyan–Magenta). The vertical axis is defined as the axis passing from the top face to the bottom face. Since there are three orthogonal face pairs, each of which can be oriented in two ways, this results in 6 ways to define the vertical axis, corresponding to 6 fundamental shape-views. Rotating each shape-view around the vertical axis yields the 4 symbolic descriptions associated to the shape. The starting direction of the symbolic description is the nearest cube (colored in blue in the figure).

**Dataset of Module I.** For each object, 250 views are uniformly sampled from the sphere, resulting in 50,000 images for the training set, 7500 for the validation set, and 14,500 for the testing set.

**Dataset of Module II.** For each symbolic description, we uniformly sampled 42 views within each quadrant, resulting in 201,600 images-symbolic descriptions for the training set, 30,240 for the validation set, and 58,464 for the testing set.

---

[6]Available at github.com/aleksandr-krylov/shepard-metzler-shape-renderer

**Dataset of Module III.** For each orthographic shape-view, we sample 4 target views uniformly on the circle, we then create the source views by taking the target poses and adding the disparity angle (4 possibilities). We do that for both 'same trials' and 'mirror trials', resulting in 38,400 trials for the training set, 5760 for the validation set, and 11,136 for the testing set (or the set on which we tested our model on the mental rotation task, see Section 4.2 and Section 5).

### E.3. Additional Results and Analyses

**We attempted to assess the correlation between human reaction times and action counts across trials.** We observe that human subjects tend to take more actions in the 180° than in the 120° condition. We first hypothesized that this larger number of actions could be responsible for the larger reaction times of humans at 180° than at 120°. However, our analysis of human data revealed no correlation between reaction times and number of actions across trials for a fixed angular disparity (App. Fig. 13), invalidating our hypothesis. Intriguingly, in the Action setup, humans were as fast in the 180° as in the 120° condition (Fig. 2.A). We thus hypothesize that the neural mechanism for the rotation of mental representations in the brain—presumably taking place in the No-Action setup but not in the Action setup, where rotations are carried out manually—may take more time for larger rotation angles, explaining the linear relation of reaction times with angular disparity in the No-Action setup between the 120° and 180° conditions. While our model does not explicitly propose a neural mechanism for the rotation operation in latent space, we propose in App. B. Additional Discussion a biologically plausible neural mechanism which does exhibit a linear relation between angle of rotation and time.

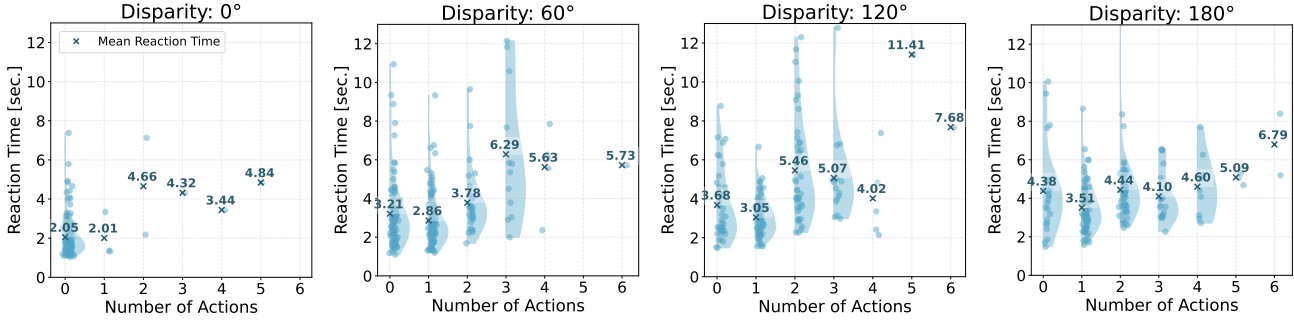

*Figure 13.* Reaction times as a function of number of actions per trial across angular disparities (from left to right: $0°, 60°, 120°,$ and $180°$). Each scatter point shows an individual successful match trial—i.e., a trial involving pairs of the same object where the candidate correctly identified them as 'same'. Distributions of reaction time are shown for numbers of actions with more than five trials only.

**We attempted to predict human behavior at a single-trial resolution.** We next sought to model human behavior on a trial-by-trial basis (e.g., as in Stewart et al., 2022), where a trial corresponds to a specific pair of shapes shown at specific angles. However, our analysis of human actions showed poor consistency in the number of actions taken for a given trial across individuals, preventing us from attempting to predict the number of actions at a single trial resolution (App. Fig. 14). We also observed poor consistency of reaction times for a given trial across individuals, preventing us from modeling reaction times at a single trial resolution (App. Fig. 15). We thus focused on modeling average behavior across trials and individuals.

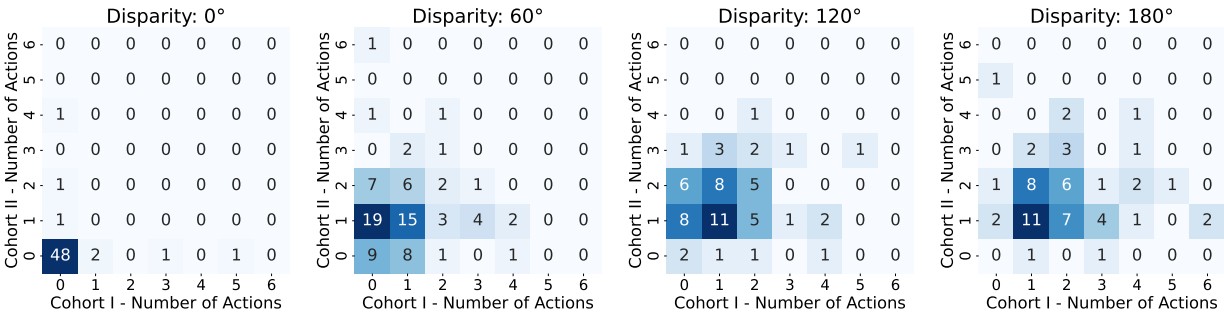

*Figure 14.* Confusion matrices of number of actions between Cohort I and II for successful match trials across angular disparities (from left to right: $0°, 60°, 120°,$ and $180°$). Except for the $0°$ condition, where subjects almost never take an action, the number of actions taken for the same task varies inconsistently between cohorts.

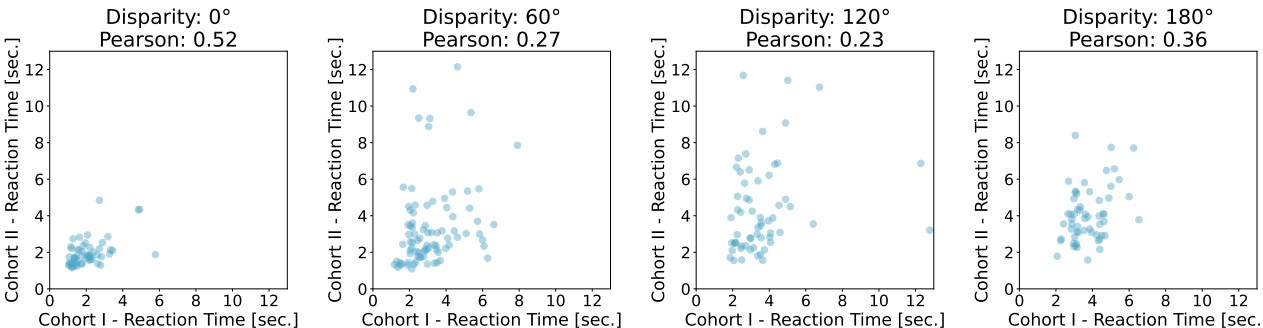

*Figure 15.* Scatter plot showing reaction time correlation between Cohort I and II for successful match trials in the Action Setup across angular disparities (from left to right: $0°, 60°, 120°,$ and $180°$). The reaction time Pearson correlation between cohorts I and II across trials is: $0.52$ for condition $0°$, $0.27$ for condition $60°$, $0.23$ for condition $120°$, and $0.36$ for condition $180°$. Condition $0°$ exhibits moderate correlation in reaction time between cohorts. The lower correlations in other conditions indicate greater variability in reaction time between cohorts.

**We attempted to compare error consistency between human cohorts.** We sought to examine whether two cohorts of humans, who were exposed to the exact same series of trials, made the same mistakes, i.e., whether both responded 'mismatch' when it was a 'match' or vice versa. This was done using the trial-by-trial error consistency method proposed by Geirhos et al. (2020), which is a metric that measures if two decision makers (a decision maker can be either a human or an algorithm) share the same strategy by asking: given their error rates, did both cohorts make the same mistakes more often than expected by chance? In other words, if the two decision makers were responding randomly, some of their incorrect responses would coincide randomly, the error consistency quantify this chance level: an error consistency of $1.0$ indicates perfect agreement of mistakes beyond chance, a value of $0.0$ indicates agreement by chance and a value $< 0.0$ indicates agreement worse than chance. The error consistency is an adaptation of Cohen's $\kappa$, where, instead of using the raw responses of two observers $i$ and $j$, we use their correctness:

$$\kappa_{i,j} = \frac{c_{obs_{i,j}} - c_{exp_{i,j}}}{1 - c_{exp_{i,j}}}, \tag{3}$$

where for $n$ trials, the observed error overlap is $c_{obs_{i,j}} = \frac{e_{i,j}}{n}$ with $e_{i,j}$ being the number of time observers were either correct or incorrect, and the expect overlap due to chance is $c_{exp_{i,j}} = p_i p_j + (1 - p_i)(1 - p_j)$, with $p_i$ and $p_j$ denoting the accuracies of observers $i$ and $j$, respectively. Over 550 trials in the action setup, we found that cohort1-to-cohort2 showed an error consistency of $0.163$, with both cohorts having high accuracy on the trials (respectively $96.18\%$ and $94.73\%$). As for the model, it has an accuracy of $96.18\%$, and the error consistency model-to-cohort1 is $0.01$ and model-to-cohort2 is $0.08$. In conclusion, we observe a weak, but non-negligible, agreement beyond chance between human cohorts, and that the model does not capture humans' errors.

## E.4. Additional Ablation Details

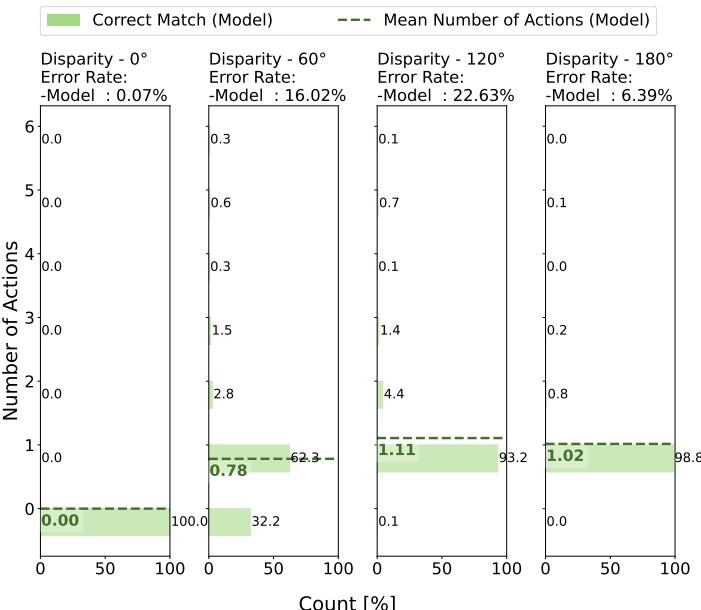

*Figure 16.* Distribution of number of actions taken per trial in the symbolic module ablation where the VSM encoder is kept, across angular disparities $(0°, 60°, 120°,$ and $180°)$. Each panel shows the normalized distribution of action count per successful match trials, along with the error rate and average number of actions taken by the ablated Model.

### E.4.1. SIAMESE NETWORKS

**We tested a classical Siamese architecture on the task.** Each Siamese encoder took as input one of the object view (i.e., an image). The output of the Siamese encoders were then concatenated and fed to a classifier on the task of predicting whether the two objects are the same or different (binary classifier). This feed-forward architecture comes most naturally to the ML practitioner for this type of same/different task, and it has indeed been used in similar tasks before (e.g., Ricci et al. 2018). For the Siamese encoder backbone, we tested a ResNet-18, a ResNet-34, a ResNet-50, a ViT-B-16, and a SL-ViT (App. Fig. 17). When the Siamese encoder backbone was of the ResNet family, the last block of the ResNet was used as part of the classifier, and was fed the concatenated outputs of the siamese networks. For ViT backbones, the concatenated CLS tokens were directly fed to a 1-hidden layer classifier. The whole architecture was trained end-to-end on the task of predicting whether the two objects are the same or different. In all cases, we found that these models were unable to perform the task. On training objects, the models were close to 100% accuracy, even on views unseen during training, but on test objects (objects not seen during training), the performance systematically fell to chance levels. Interestingly, these architectures were able to perform the task accurately when the object rotation was confined to the plane of the image (i.e., a 2D rotation). This behavior fundamentally differs from humans', who compare with the same facility objects rotated in the plane and in depth (Shepard & Metzler, 1971).

The Siamese networks were trained with the following optimizer and parameters:

- optimizer: SGD

- lr: 0.01, momentum: 0.9

- scheduler with number of warm-up epochs: 5, warm-up decay: 0.01

On the in-plane rotation task, the siamese networks were trained on 1024 training pairs for 100 epochs. On the in-depth rotation task, they were trained on 2048 training pairs of Shepard-Metzler shapes for 100 epochs. They were then tested on unseen Shepard-Metzler shapes.

For the ResNet-family encoders, the last ResNet block was removed from the siamese encoders and fed as an input the concatenated outputs of the siamese networks. The output of the last ResNet block was then fed into a 1 hidden layer

classifier (256 units => ReLU => 1 output unit => sigmoid) trained on the binary label (match / mismatch) with a cross-entropy loss. For the ViT-family encoders, the concatenated CLS tokens were directly fed to the one-hidden-layer classifier. None of these architectures were able to learn the task for in-depth rotation.

We also attempted a curriculum learning approach for the ResNet-18 siamese networks (App. Fig. 17D), by progressively extending the range of angular difference visited. For each range, we trained the networks on 512 samples for 100 epochs. The first range was -20° to 20° and the last one -180° to 180°. The siamese networks still failed the task on test shapes at large angles.

We also tried to replace the supervised loss with a supervised contrastive loss (Khosla et al., 2020) without success (data not shown).

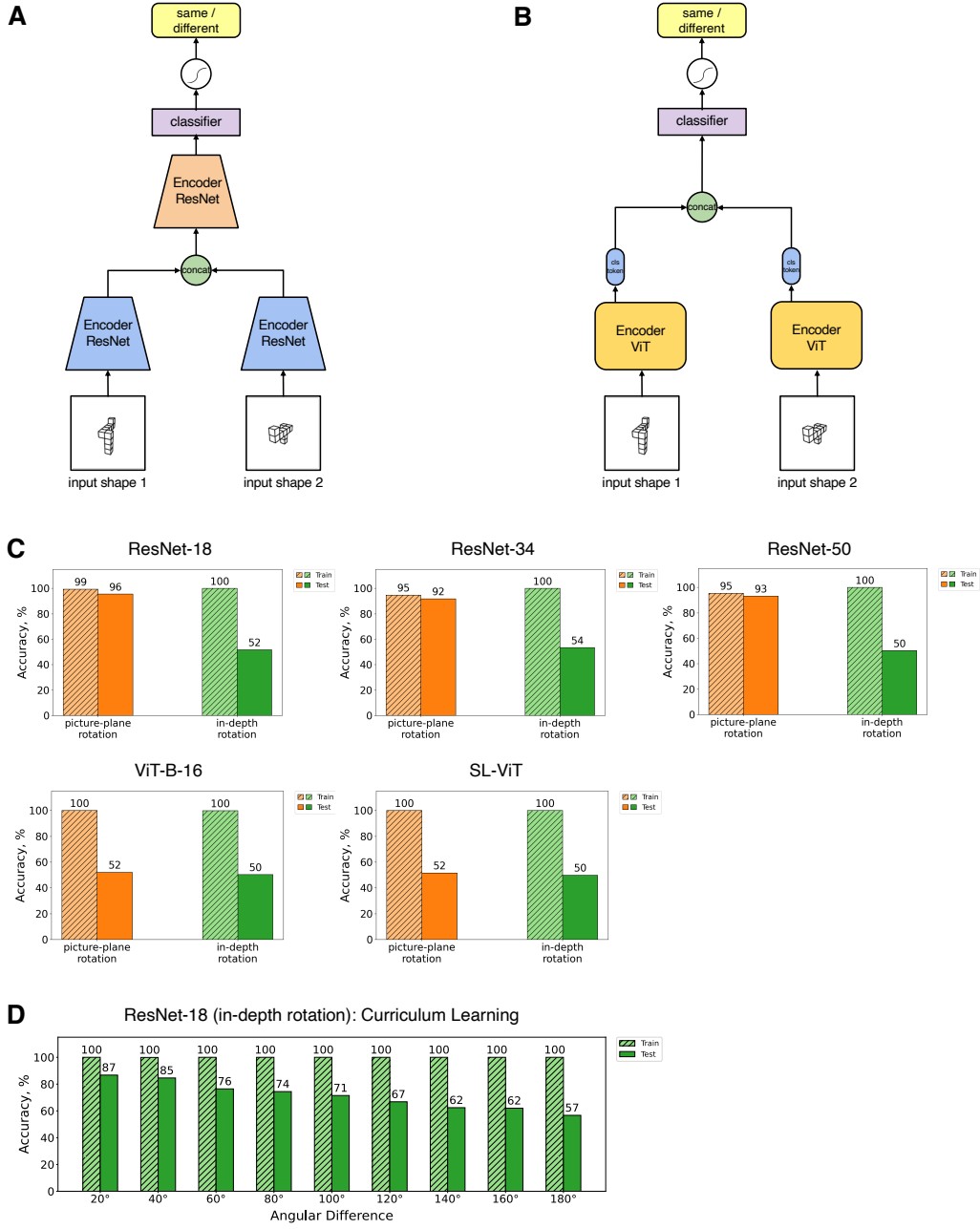

Figure 17. Siamese networks. (A) With ResNet backbones. (B) With ViT backbones. (C) Results on training and testing sets, for 2D and 3D rotation. (D) Result obtained through curriculum learning (small angles of rotations learned first).

