# OpenReview forum: "A Deep Learning Model of Mental Rotation Informed by Interactive VR Experiments"
_ICML.cc/2026/Conference — ICML 2026 regular_

### Official Review · Reviewer_aiby · 2026-03-11

**Soundness:** 4
**Presentation:** 3
**Significance:** 2
**Originality:** 2
**Overall Recommendation:** 5
**Confidence:** 3

**Summary:**

The authors present a three-part mechanistic model for how humans perform mental rotation tasks. The model uses an equivariant encoder, an object encoder, and a discriminating MLP. Accompanying the model is a novel virtual-reality mental rotation dataset. Qualitative comparisons are made between the model and the human performance. Several ablations are done on the different model parts to establish their contributions.

**Compliance With Llm Reviewing Policy:**

Affirmed.

**Final Justification:**

I believe the authors have done a very thorough job, both in the initial submission and during their rebuttals. This was acknowledged by the other reviewers and me. Therefore, I have raised my score.

**Key Questions For Authors:**

Are the latent rotations simply constrained affine transformations?
Do i correctly understand the modules are trained sequentially? Is there a reason for this?

**Limitations:**

See weaknesses above.

**Strengths And Weaknesses:**

**Strengths**
- The newly collected dataset is interesting.
- There are extensive ablation studies as well as tests for architectural invariances (e.g. Resnet vs ViT for module 2)
- There is a proper treatment of the existing literature on mental rotation.
- Behavioural findings unexplained by the model are nicely presented and discussed.

**Weaknesses**
- While I think the quality of the work is high, I worry it may only be relevant to a small community, even within the cognitive sciences.
- I find the presentation of Figute 6 to be odd. Human vs model comparison should be more directly visible.
- I would like to see ablation results in the main text. Especially if the claim is this is a mechanistic model, it’s very important to directly see each module’s contribution.

---

> ### Author Rebuttal · Authors · 2026-03-30
>
> We thank the reviewer for their careful review, and for complimenting our work by stating that “the newly collected dataset is interesting”, and for appreciating “the extensive ablation studies” and “the proper treatment of the existing literature on mental rotation”.
>
> **W1: While I think the quality of the work is high, I worry it may only be relevant to a small community, even within the cognitive sciences.**
> We appreciate the kind words regarding the quality of the work, and for the opportunity to clarify its broader relevance. This work attempts to understand how humans perform mental rotation, by asking: *“What kinds of representations need to be enforced for human-like spatial reasoning, such as mental rotation, to emerge in deep learning models? Should the representations be explicit 3D geometric embedding, possibly equivariant to transformations, or more symbolic and abstract representations that support reasoning about composite parts, occlusion, and object relations—or some mixture of both?”* We show that both are needed: 3D equivariant representation to support rotation, and a symbolic representation to support similarity judgment and rotations choice. We are aware that the answer we are providing serves as a building block, rather than a final answer to spatial reasoning human-level ability. Nevertheless, recent deep learning approaches – multimodal large language models (Yang et al., 2025), large vision models (Kosoy et al., 2025) – still struggle on spatial reasoning tasks. As for video language models, although they perform surprisingly well on vision tasks they were not trained on, mental rotation remains stubbornly hard for them (Wiedemer et al., 2025). Our work suggests that explicitly enforcing the right representational inductive biases, rather than hoping they emerge from scale, may be a promising direction for closing this gap.
>
> References:
> - Yang, J., Yang, S., Gupta, A. W., Han, R., Fei-Fei, L., and Xie, S. Thinking in space: How multimodal large language models see, remember, and recall spaces. *In Conference on Computer Vision and Pattern Recognition (CVPR)*, 2025.
> - Kosoy, E., Dahmani, A., Lampinen, A. K., Comsa, I. M., Jeong, S., Dasgupta, I., and Allen, K. Decoupling the components of geometric understanding in vision language models. *Transactions on Machine Learning Research (TMLR)*, 2025.
> - Wiedemer, T., Li, Y., Vicol, P., Gu, S.S., Matarese, N., Swersky, K., Kim, B., Jaini, P. and Geirhos, R. Video models are zero-shot learners and reasoners. *arXiv preprint arXiv:2509.20328*, 2025.
>
> **W2: I find the presentation of Figute 6 to be odd. Human vs model comparison should be more directly visible.**
> Figure 6 presents human and model action counts per disparity condition (0°, 60°, 120° and 180°), in two separate bar plots. The human bar plot is on the right in blue, and the model one on the left in purple, allowing a direct side-by-side comparison. Moreover, we are interested in making the presentation clearer, so we welcome any specific suggestions and will improve the figure in the camera-ready version accordingly.
>
> **W3: I would like to see ablation results in the main text. Especially if the claim is this is a mechanistic model, it’s very important to directly see each module’s contribution.**
> Ablation results are presented in the main text (in Section 5.1. Ablations). Nevertheless, we moved to Appendix E.4 the detailed description of the different Siamese architectures used in the first ablation, but as none worked on the evaluated task (in-depth rotation), as indicated by their accuracy close to chance-level on the test objects, we mainly report that they fail to solve the task for in-depth rotation. Is the reviewer referring particularly to this Appendix E.4?
>
> **Q1: Are the latent rotations simply constrained affine transformations? Do i correctly understand the modules are trained sequentially? Is there a reason for this?**
> Regarding the latent rotations: Indeed, to rotate the latent space by a desired angle, we simply multiply it by a rotation matrix, which is an affine transformation constrained to the group of 3D rotation SO(3). Regarding sequential training, please see our response to ***Reviewer YGnC - Q1***.

---

> > ### Author Rebuttal · Reviewer_aiby · 2026-03-31
> >
> > I thank the authors for their responses. I am now more convinced that the findings of this paper are relevant to a larger community. I suggest that the authors provide such a motivation in the paper as well.
> >
> >  Regarding Figure 6, I meant that it'd be easier for the reader if the model and human behaviours were shown side by side in the same subplots, rather than being faceted into two different subplots. e.g. blue bars next to purple bars

---

> > > ### Author Response · Authors · 2026-04-06
> > >
> > > We sincerely thank the reviewer for their time and for raising their score. We are glad their concerns have been addressed and appreciate their suggestion to better motivate the broader relevance of our work. We will incorporate this in the revision.

---

### Official Review · Reviewer_t5az · 2026-03-13

**Soundness:** 3
**Presentation:** 3
**Significance:** 2
**Originality:** 3
**Overall Recommendation:** 4
**Confidence:** 3

**Summary:**

This paper proposes a mechanistic model of human mental rotation, which consists of a neural encoder (convolutional auto-encoder), a neuro-symbolic object encoder ( ViT), and a neural decision agent (MLP). The neural decision agent can output either a similarity decision or a rotation action to take. By taking the rotation action, the model simplifies subsequent similarity judgments. The model evaluation shows comparable accuracy to humans, and the model’s action count is analogous to thumbstick actions in human VR experiments.

**Compliance With Llm Reviewing Policy:**

Affirmed.

**Final Justification:**

My concerns are addressed in the rebuttal. I am raising the overall recommendation to 4.

**Key Questions For Authors:**

Since training the MLP to directly predict same or different objects has comparable accuracy to the full unablated model. Does that mean the action prediction in module 3 is an interpretable but functionally redundant component? Why is this mechanistically necessary?

**Limitations:**

yes

**Strengths And Weaknesses:**

**Strength**:

1. This paper recreated  Shepard & Metzler mental rotation experiment with an interactive VR human experiment, and evaluated actions taken by humans in an interactive experiement instaed to reaction time. This circumvents the hypothesis of a fixed and continuous mental rotation speed.
2. Model components are ablated to evaluate the individual contribution of each module.
3. The paper is well written and easy to follow.

**Weakness**:

1. The model formulates mental rotation as discrete categorical decisions, whereas human mental rotation is more like a continuous process. This gap makes the mechanistic interpretation unclear, whether it captures the same underlying form of mental rotation used by humans.
2. In the interactive setting, human actions are reflected as joystick operations in VR, which is a noisy proxy for the internal cognitive process. Motor execution is noisy, and humans may operate multiple times even for a single intended rotation angle. Moreover, larger intended rotation angles may naturally require longer or more segmented physical manipulation. As a result, it is unclear if correlations in action counts between humans and the model can be interpreted as evidence for a shared mental rotation mechanism.

---

> ### Author Rebuttal · Authors · 2026-03-30
>
> We thank the reviewer for their careful review, and for complimenting our work by stating that “the paper is well written and easy to follow”.
>
> **W1: The model formulates mental rotation as discrete categorical decisions, whereas human mental rotation is more like a continuous process. This gap makes the mechanistic interpretation unclear, ...**
> We note that our model accommodates both continuous and discrete aspects of mental rotation: the equivariant component supports continuous rotation, while the symbolic one supports similarity judgment and action decision. This hybrid structure is consistent with reports of both continuous and non-continuous rotation strategies found in the literature and observed in our experiments (see Discussion section paragraph *"Symbolic or Spatial representations? We Answer ‘Both!’"*). Our Quadrant Hypothesis is introduced explicitly as a plausible empirically-based explanation, and the fact that it recapitulates the human behavioral patterns does not prove its uniqueness. In the hypothesis, we propose 4 symmetrical reference frames of 90° as the simplest explanation consistent with the behavioral data: few actions taken and humans positioning objects within ±45° before judgment. The core idea of the quadrant hypothesis is not limited to Shepard-Metzler objects, as any 3D object can be placed in a bounding cube: if we fix the axis of rotation, the symbolic description would correspond to an abstraction of the information visible in each of the 4 vertical faces. Our choice of 4 categories is motivated by the ±45° empirical observation, though other discrete strategies cannot be ruled out from the few actions observations. Moreover, the quadrant-based symbolic encoding is geometrically principled for reasoning about object relationships (see response to ***Reviewer 8wFz - W1.1***). Finally, Shepard–Metzler objects are convenient as they are already composed of cubes, allowing a natural symbolic code as the relative directions between the component cubes.
>
> **W2. In the interactive setting, human actions are reflected as joystick operations in VR, which is a noisy proxy for the internal cognitive process. Motor execution is noisy, ...**
> This is a really good remark, but actually, our model does not really capture the noise responsible for distribution of actions, it models the rough increase of number of actions with angular disparity at a rather coarse level. Moreover, action counts in our interactive setting are low on average: 0 for 0°, 0.97 for 60°, 1.41 for 120°, and 1.71 for 180°.  We also observe that humans exhibit greater variability in the number of actions across conditions, and we attribute this to the inherent noise in human interaction. It would be interesting to add to our model a notion of noise due for instance to the necessity of segmenting long physical manipulations. We leave this for future work.
>
> **Q1: Since training the MLP to directly predict same or different objects has comparable accuracy to the full unablated model. Does that mean the action prediction in module 3 is an interpretable but ...**
> The last ablation involves assessing whether the mental rotation task can be solved, using for Module 3, a MLP without action decision outputs. As a reminder, the standard Module 3 used in the model, is a MLP composed of 3-layers trained on 5 classes: 2 similarity decisions (same, mirror) and 3 rotation actions (90°, -90°, 180°). Training Module 3 on ‘similarity decisions’ only, hereafter the 2-class version, achieves 97.03% accuracy, comparable with the original 5-class version of Module 3 (96.13%). This 2-class version, however, fails to recapitulate human behavior as it takes no actions by design. Now, reducing Module 3 from 3-layers to 1-layer, and re-evaluating a 5-class and 2-class version reveals an advantage for the 5-class: the 5-class version achieves 96.05% while still replicating human action counts (comparable to its 3-layers counterpart), whereas the 2-class version drops to 86.23% accuracy on the mental rotation task. We agree with the reviewer that accuracy alone provides no incentive to prefer the 5-class over the 2-class version; a sufficiently large MLP trained solely on similarity decisions achieves comparable accuracy. However, it is important to be careful with the usage of the word 'necessity' in our context: as this paper presents a mechanistic model of mental rotation, we argue that ‘necessity’ should be evaluated against richer criteria than accuracy alone. Our model's action decisions, not only recapitulate the parsimonious number of actions observed in humans, it allows us to recover the relations between objects to simulate rotation; information that is lost by a rotation-invariant detector. A module trained on similarity decisions only will fail to capture those relations. We do discuss that in Appendix B (paragraph *'Mental Actions: Why and How?'*), and will move this paragraph in the main text for the camera-ready version.

---

> > ### Author Rebuttal · Reviewer_t5az · 2026-04-03
> >
> > Thank the authors for the detailed response. I will raise my score to 4.

---

> > > ### Author Response · Authors · 2026-04-06
> > >
> > > We sincerely thank the reviewer for their time, for confirming that their concerns have been addressed and for raising their score. We look forward to the revised score.

---

### Official Review · Reviewer_8wFz · 2026-03-13

**Soundness:** 2
**Presentation:** 2
**Significance:** 3
**Originality:** 3
**Overall Recommendation:** 4
**Confidence:** 4

**Summary:**

This paper investigates human mental rotation—the ability to compare objects viewed from different perspectives—through virtual reality (VR) experiments. Inspired by the experimental findings, the authors design a model that first recovers a manipulable 3D spatial representation from a single 2D image using an equivariant neural encoder, then converts this spatial representation into a symbolic object description via a neuro-symbolic object encoder, and finally uses an MLP to make similarity judgments and rotation action decisions based on paired symbolic descriptions. The paper claims that the model can fit human accuracy, reaction times, and action behavior.

**Compliance With Llm Reviewing Policy:**

Affirmed.

**Final Justification:**

This paper addresses the meaningful problem of human mental rotation and attempts to explain humans’ internal cognitive strategies by combining an interactive VR experiment with a mechanistic model. The topic has a certain degree of originality and potential significance. The experimental paradigm is well designed and thought-provoking, the model construction is aligned with the paper’s core hypothesis, and the figures are clear and visually appealing. These are all strengths of the work.

\
That said, my main reservations about the paper still remain. First, in terms of soundness, the proposed “quadrant-based symbolic encoding” can be viewed as a plausible explanation, but the current evidence is still insufficient to show that it provides strong support for, or a unique explanation of, the mechanism underlying human mental rotation. Other discrete strategies or heuristic mechanisms could also account for the observed phenomena. Second, in terms of experimental support, the empirical evidence is still relatively weak. In addition, the task setup is relatively simple, which limits the extent to which the current results support the model’s true capability and generalization. Finally, in terms of clarity, although the rebuttal clarified some issues, it also indirectly suggests that the initial submission was indeed not sufficiently clear in its writing and presentation of results, and still requires substantial improvement.

\
The authors’ rebuttal provided helpful explanations for some of these issues and clarified several aspects of the experimental setup and modeling motivation. As a result, my evaluation of the paper has improved compared with my initial assessment, and I have raised my score accordingly. Overall, however, the rebuttal mainly shifted some concerns from being “unclear” to being “understandable,” rather than fundamentally resolving my core concerns about the strength of the mechanistic claim, the adequacy of the experiments, and the reliability of the conclusions. Therefore, my final evaluation is more positive than before.

\
Overall, I appreciate the research direction, the value of the problem, and the degree of novelty in this work, and I thank the authors for their careful response to the reviewers’ comments. However, in terms of rigor, experimental persuasiveness, and overall maturity, I believe the paper still has clear room for improvement.

**Key Questions For Authors:**

1. The paper’s central contribution is to propose a mechanistic explanation of human mental rotation. However, the VR finding of “few actions + coarse alignment” does not uniquely support the authors’ quadrant-based symbolic encoding. Other forms of discrete strategies, coarse-grained pose binning, or learned heuristic decision boundaries could also explain similar behavior. At present, this is better viewed as a plausible explanation rather than one that is strongly confirmed by the data.
2. The model is not compared against other baselines.
3. When comparing the model with humans, the reported results are given along different dimensions and are difficult to align. The model results are presented in terms of match/mismatch, whereas the human results are presented in terms of Action/No-Action.
4. There are only 15 valid participants, which is too small a sample size and weakens the reliability of the conclusions.
5. The task setup may be overly simplistic. Both humans and the model can solve most cases with just a single action, and the success rate is already around 95%, suggesting possible overfitting to the dataset. Under such conditions, it is difficult to determine whether the observed performance gains come from handling corner cases better or from a genuine improvement in core capability. This may affect the reliability of the paper’s conclusions.
6. The evidence for generalization is insufficient. The current model and experiments are limited to Shepard–Metzler objects, fixed elevation, specific viewpoint changes, and a particular symbolic encoding scheme. There is no evidence showing whether the framework can generalize to other 3D。objects, other rotation axes, different viewing conditions, or more general spatial reasoning tasks.
7. There are many qualitative results but relatively few quantitative results.
8. Some ablation studies do not establish the necessity of the corresponding modules, especially the last ablation experiment.
9. In Figure 2(b), the 180-degree result is left-right asymmetric, whereas the other angles are symmetric. The reason for this discrepancy should be analyzed in more detail.

**Limitations:**

yes

**Strengths And Weaknesses:**

Strengths:
1.	It studies a meaningful problem and may help the community better understand human mental rotation ability.
2.	It introduces an interactive VR task and infers internal cognitive strategies by observing participants’ actual rotation behavior.
3.	The model design is aligned with the paper’s core hypothesis.
4.	The figures are visually appealing, and the quality of visualization is high.
Weaknesses:
1. Model
1.1. It is intuitive that humans may compare 3D objects by quadrant, but the further step of converting them into symbolic representations is debatable. Symbolic sequences are less intuitive than images, and humans may not actually process the task in this manner.
1.2. The paper’s central contribution is to propose a mechanistic explanation of human mental rotation. However, the VR finding of “few actions + coarse alignment” does not uniquely support the authors’ quadrant-based symbolic encoding. Other forms of discrete strategies, coarse-grained pose binning, or learned heuristic decision boundaries could also explain similar behavior. At present, this is better viewed as a plausible explanation rather than one that is strongly confirmed by the data.
2. Experiments
2.1. The model is not compared against other baselines.
2.2. When comparing the model with humans, the reported results are given along different dimensions and are difficult to align. The model results are presented in terms of match/mismatch, whereas the human results are presented in terms of Action/No-Action.
2.3. There are only 15 valid participants, which is too small a sample size and weakens the reliability of the conclusions.
2.4. The task setup may be overly simplistic. Both humans and the model can solve most cases with just a single action, and the success rate is already around 95%, suggesting possible overfitting to the dataset. Under such conditions, it is difficult to determine whether the observed performance gains come from handling corner cases better or from a genuine improvement in core capability. This may affect the reliability of the paper’s conclusions.
2.5. The evidence for generalization is insufficient. The current model and experiments are limited to Shepard–Metzler objects, fixed elevation, specific viewpoint changes, and a particular symbolic encoding scheme. There is no evidence showing whether the framework can generalize to other 3D。objects, other rotation axes, different viewing conditions, or more general spatial reasoning tasks.
2.6. There are many qualitative results but relatively few quantitative results.
2.7. Some ablation studies do not establish the necessity of the corresponding modules, especially the last ablation experiment.
2.8. In Figure 2(b), the 180-degree result is left-right asymmetric, whereas the other angles are symmetric. The reason for this discrepancy should be analyzed in more detail.
2.9. The statement “a single action corresponding to an average rotation angle of 73.1◦ (Fig. 6.A)” is not clearly supported by Figure 6A; I could not find the corresponding information there.
3. Writing
3.1. For the main experiments and ablation studies, more results should be presented in table form, which would make performance comparisons more direct and easier to follow. At present, the paper relies too heavily on textual description, which makes the reading experience less clear.

---

> ### Author Rebuttal · Authors · 2026-03-30
>
> We thank the reviewer for their careful review, and for complimenting our work as studying “a meaningful problem which may help the community better understand human mental rotation”, and stating that the “quality of visualizations is high”.
>
> W1.1. We agree the exact biological plausibility of the symbolic code is debatable, we address this in the paper: *‘While humans may not use this exact coding scheme, it provides a tractable abstraction…’* The key insight is not the code's specific form but its granularity: compressed enough to support similarity judgment while discarding irrelevant information. This is precisely where an equivariant representation falls short: it is too fine-grained, requiring near-perfect alignment for similarity. We needed a symbolic scheme robust enough to abstract object poses to quadrant membership. Our coding scheme captures this tolerance naturally: it is invariant within a quadrant but sensitive across them, preserving the information necessary for same/mirror discrimination and action selection.
>
> W1.2 (Q1). Please see our response to ***Reviewer t5az - W1***
>
> ---
> W2.1 (Q2). One challenge in mental rotation is that humans solve the task equally well in-plane and in-depth, by reasoning about occluded object parts — a capability standard vision models lack. In Ablation 1, we compare our model against Siamese networks with standard vision encoders (ResNet or ViT) trained directly on same/mirror task; these fail on in-depth rotation, serving as a baseline. Module-wise ablations provide additional baselines within our own framework. We point to the 'Novelty' paragraph in the Discussion that situates our work within the literature and explains why direct comparison is difficult: to our knowledge, this is the first attempt at a mechanistic model of human mental rotation.
>
> W2.2 (Q3). The mental rotation task is a binary task: given a pair of objects, the goal is to answer 'match' (same object) or 'mismatch' (mirror object). Humans performed the task in two setups: one where interaction is allowed  (Action setup), and a non-interactive one (No-Action setup) – the nomination ‘Action’ and ‘No-Action’ are experimental labels. We report overall and class-wise accuracy (match and mismatch separately) for the model and both human setups.
>
> W2.3 (Q4). We recognize that a sample size of 15 participants may seem modest. Also, the model is deterministic at inference, so statistical comparison of model to human action counts is not applicable. Moreover, of 19 participants, 4 were excluded for atypical behavior using the interquartile range method (see Appendix C), leaving 15 participants each contributing for 100 trials per setup. Finally, our model design is guided not only by our experimental observations, but also by the extensive literature on mental rotation.
>
> W2.4 (Q5). Mental rotation is well-studied and reliably solved by healthy adults, with near-perfect performance given sufficient time. Humans perform equally when interactive rotation is allowed (Action) and when disabled (No-Action), and the literature supports that both setups engage the same underlying process. Regarding model overfitting: the full model (Modules 1–3) is trained on Shepard-Metzler objects but evaluated on held-out objects unseen in any orientation during training. Moreover, the model is not trained on solving the task  sequentially: Module 3 is trained on a balanced 5-class problem — predicting a similarity decision (same/mirror) or a rotation action (90°, -90°, 180°). Sequential rotation action-taking at inference is emergent. Finally, Ablation 1 shows that standard vision models struggle on in-depth rotation, directly contradicting the claim of trivial simplicity.
>
> W2.5 (Q6). Please see ***Reviewer YGnC - W1***
>
> W2.6 (Q7). We report accuracy (same/mirror) across all conditions and ablations, and quantify mean action counts per disparity condition to assess behavioral correspondence with humans. We also conduct error consistency analysis (Appendix E.3). Together, these provide a comprehensive quantitative picture of the model's behavior. We agree that the reaction time profile analysis is qualitative. Finally, we welcome any concrete suggestions for additional measures.
>
> W2.7 (Q8). Please see ***Reviewer t5az - Q1***
>
> W2.8 (Q9).  We argue that the left-right asymmetry in the 180° case reflects the shortest-path strategy employed by humans: for disparities below 180°, a clear shorter path exists and humans tend to take it, resulting in the symmetric distributions observed for 60° and 120° (Fig 2b). Disparity of 180° is the edge case where both paths are equivalent: humans default to a consistent directional preference, producing the observed asymmetry.
>
> W2.9. We thank the reviewer for spotting this. We will remove the figure reference in the camera-ready version.
>
> ---
> W3.1. We thank the reviewer for this suggestion. We will add a table to make these comparisons more visually immediate in the camera-ready version.

---

> > ### Author Rebuttal · Reviewer_8wFz · 2026-04-03
> >
> > The initial submission had too many problems, indicating that the writing still needs substantial improvement and that many issues were not clearly explained.
> >
> > The rebuttal clarified some of these concerns, so I will raise my score by one point and give a more positive evaluation.
> >
> > I hope the authors will incorporate the relevant suggestions into the revised version.

---

> > > ### Author Response · Authors · 2026-04-06
> > >
> > > We sincerely thank the reviewer for their time, for confirming that some of their concerns have been addressed, and for raising their score. We will revise carefully to improve the clarity and presentation of the paper.

---

### Official Review · Reviewer_YGnC · 2026-03-13

**Soundness:** 3
**Presentation:** 4
**Significance:** 3
**Originality:** 3
**Overall Recommendation:** 5
**Confidence:** 3

**Summary:**

This paper proposes a mechanistic deep learning model for human mental rotation, aiming not just to solve the Shepard-Metzler task but to account for behavioral signatures such as action patterns and reaction times. The core model has three stages: an equivariant neural renderer that builds a manipulable 3D latent from a single image, a vision-symbolic model that converts this latent into a sequential symbolic description, and an MLP-based decision module that either predicts match/mismatch or selects a discrete rotation action. The modeling choices are motivated by both prior literature and new VR experiments, where participants could manually rotate one object using a controller. From these experiments, the authors argue that humans often rely on a small number of ballistic, discrete actions and do not require precise final alignment before judging similarity, which motivates their “quadrant-dependent symbolic representation” hypothesis. Empirically, the model achieves high accuracy on held-out objects, roughly matches the human distribution of action counts, and partially explains the classic increase of response time with angular disparity, though it does not explain the gap between 120° and 180° conditions.

**Compliance With Llm Reviewing Policy:**

Affirmed.

**Final Justification:**

This paper proposes a mechanistic deep learning model of human mental rotation, integrating equivariant, neuro-symbolic, and recurrent components to capture behavioral patterns from both classical experiments and a novel VR setup. The framework is well-motivated by cognitive literature, and the realization of these ideas into a working model is a noteworthy contribution. The ablation studies are carefully designed and convincingly justify each module's role.

My primary concern was the narrow scope—only Shepard–Metzler objects and Y-axis rotation—which limits the generality of the mechanistic claim. The authors acknowledged this openly, positioning the work as a proof-of-concept and noting that the equivariant renderer supports full SO(3) and the quadrant hypothesis extends to other axes. I found this response adequate; the limitation is real but does not undermine the paper's core contribution.

On the modular training question, the authors clarified that each module has a distinct training objective and that end-to-end training was attempted but proved computationally prohibitive. No overclaims about neural plausibility were made, which I appreciate.
The rebuttal reinforced my prior assessment. I maintain my score of 5 (Accept).

**Key Questions For Authors:**

1. Why are the three modules trained independently rather than end-to-end? Is this a matter of engineering convenience, or is there a cognitive rationale? Is there evidence that the human brain develops or organizes these capacities in a similarly modular fashion?

**Limitations:**

yes

**Strengths And Weaknesses:**

### Strength
**Novel approach to implementing mental simulation.** The paper translates a series of cognitive psychology insights into an ML framework: Module I (adapted from prior work) learns 3D representations from 2D images; Module II converts these 3D representations into symbolic descriptions; and Module III treats objects with different views as equivalent when their symbolic representations match. It is impressive to see these cognitively inspired ideas realized as a working model. While each individual component follows previously established directions, the full framework—showing that visually distinct views can be symbolically equated—is a meaningful contribution to the ML community as well.

**Concrete ablation study.** The authors carefully ablate each module to justify why that specific component is necessary, effectively demonstrating the robustness of the overall model design.

### Weakness
**Limited dataset and experimental setup.** The model operates exclusively on Shepard–Metzler objects composed of 10 cubes. Human mental rotation, however, applies effortlessly to arbitrary objects—chairs, letters, faces, etc.—yet this model would require full retraining for any new object category. This makes the claim of a "mechanistic model of mental rotation" rather narrow in scope. Additionally, the experiments only involve rotations around the Y-axis (depth rotation), whereas the original Shepard & Metzler experiments addressed both picture-plane and depth rotations. Only covering one axis is a notable limitation.

---

> ### Author Rebuttal · Authors · 2026-03-30
>
> We thank the reviewer for their careful review, and for complimenting our work by stating that “it is impressive to see these cognitively inspired ideas realized as a working model” and that our framework constitutes a “meaningful contribution to the ML community”.
>
> **W1:  The model operates exclusively on Shepard–Metzler objects composed of 10 cubes. Human mental rotation, however, applies effortlessly to arbitrary objects—chairs, letters, faces, etc.—yet this model would require full retraining for any new object category. This makes the claim of a "mechanistic model of mental rotation" rather narrow in scope. Additionally, the experiments only involve rotations around the Y-axis (depth rotation), whereas the original Shepard & Metzler experiments addressed both picture-plane and depth rotations. Only covering one axis is a notable limitation.**
>
> We acknowledge generalization beyond Shepard–Metzler objects as a current limitation, addressed in Appendix B (*'Biological Plausibility of Backpropagation, Data Diet and Coding Scheme'*). However, Shepard-Metzler objects are well-suited for studying mental rotation: their chirality, or existence of distinct mirror versions, is precisely what makes same/mirror discrimination non-trivial. This property is essential to the task design and may not generalize to arbitrary 3D objects. We do not claim to have solved general spatial reasoning, but we rather present a proof-of-concept attempt, informed by the existing literature on mental rotation and our VR experimental setup. We claim that humans can solve mental rotation because they rely on an equivariant representation of the object that they can manipulate, and a symbolic representation that operates at a specific level of abstraction to support similarity assessment. If we enable those components (equivariant and symbolic) in a model, then we can solve mental rotation. Moreover, we note: (1) the equivariant renderer works for any 3D rotation, covering arbitrary rotation transformation on SO(3); (2) the fixed rotation axis mirrors our human experimental setup for model-human comparison, as for the quadrant hypothesis, it extends to other axes and spans the entire 3D rotation space (as illustrated and described in Appendix Figure 12); (3) the sequential symbolic encoding is object-specific by design as it exploits the discrete compositional structure of Shepard-Metzler object. Nevertheless, extending this framework, notably the symbolic encoding to other object classes and rotation axes is a natural and important direction for future work.
>
>
> **Q1: Why are the three modules trained independently rather than end-to-end? Is this a matter of engineering convenience, or is there a cognitive rationale? Is there evidence that the human brain develops or organizes these capacities in a similarly modular fashion?**
>
> The modules are trained sequentially, and not end-to-end. The principal reason is that each module is responsible for a specific task in the pipeline and each one has its own dedicated training procedure. Going from the end, Module 3 is trained to discriminate the quadrants relationship between pairs of symbolic representations (regardless of if they are ‘same’ or ‘mirror’ object), or if the objects are located in the same quadrant to discriminate whether they are the ‘same’ or ‘mirror’ version of each other: ultimately Module 3 is trained on 5 classes without applying any rotation during training; it uses Module 1 and 2 as feature extractor in a Siamese fashion on pairs of images. So by design Module 3 needs to be trained separately. As for Module 1 and Module 2 they could potentially be trained end-to-end together, we actually tried that and it was computationally too heavy: we needed GPUs with bigger VRAM to store all the compute during backward pass, and it was converging slowly in contrast to their separate counterpart. Regarding neural plausibility of modular organization: we make no claims about how humans develop or neurally organize these capacities, it is beyond the scope of the current work. The modular structure of our model is a computational choice.

---

> > ### Author Rebuttal · Reviewer_YGnC · 2026-04-02
> >
> > Thanks for your response. I maintain my positive score.

---

> > > ### Author Response · Authors · 2026-04-06
> > >
> > > We sincerely thank the reviewer for their time and their positive assessment and continued support.

---

### Decision · Program_Chairs · 2026-04-30

**Decision:**

Accept (regular)

**Comment:**

This paper proposes a (1) experiment, and (2) model of mental rotation in humans. In the experiment, people are asked to perform the classic Shepherd-Metzler mental rotation experiment in VR where they either have the ability to rotate one of the objects (the "Action" setup) or not (the "No-Action" setup). The paper reproduces the initial findings about accuracy and reaction times in VR, and proposes a model involving 3D reconstruction, symbolic translation of 3D into symbols, and finally an MLP for predicting, based on the symbolic descriptions of the two objects, whether an "action" should be taken to rotate an object, or a prediction about the similarity.

__All reviewers appreciated that the authors showed that this model predicts human behavior on this task very well.__ They show that ablations to the model (operating in 2D instead of 3D, not having the symbolic representation) make the prediction of human behavior much worse.

The __main criticisms are that the model is restricted in scope (it only applies to one axis of rotation, only applies to these Shepherd-Metzler type objects) and the approach may be of limited interest to the ICML community__. I agree with these concerns. In particular, the symbolic translation is unlikely to extend to other categories of objects, and the MLP for predicting rotation action direction may also fail to work as well if needing to predict an axis of rotation in addition to how much to rotate. The paper is also unlikely to be of significant interest to the ICML community despite the experimental soundness given the restrictiveness of the task.

Other concerns brought up included the soundness of the symbolic processing claim relative to the empirical findings, and some concerns about the clarity of the presentation.

I therefore recommend a "weak accept". If this paper is rejected, I would recommend submitting to a more cognitive science-oriented venue (e.g. the computational cognitive neuroscience conference, a journal, or the annual proceedings of the cognitive science society).